# Evidence for late-glacial oceanic carbon redistribution and discharge from the Pacific Southern Ocean

Shinya Iwasaki [1,2,6] ✉, Lester Lembke-Jene [3,6], Kana Nagashima[1,6], Helge W. Arz [4,6], Naomi Harada[1,5,6], Katsunori Kimoto[1,6] & Frank Lamy [3,6]

Southern Ocean deep-water circulation plays a vital role in the global carbon cycle. On geological time scales, upwelling along the Chilean margin likely contributed to the deglacial atmospheric carbon dioxide rise, but little quantitative evidence exists of carbon storage. Here, we develop an X-ray Micro-Computer-Tomography method to assess foraminiferal test dissolution as proxy for paleo-carbonate ion concentrations ($[CO_3^{2-}]$). Our subantarctic Southeast Pacific sediment core depth transect shows significant deep-water $[CO_3^{2-}]$ variations during the Last Glacial Maximum and Deglaciation (10-22 ka BP). We provide evidence for an increase in $[CO_3^{2-}]$ during the early-deglacial period (15-19 ka BP) in Lower Circumpolar Deepwater. The export of such low-carbon deep-water from the Pacific to the Atlantic contributed to significantly lowered carbon storage within the Southern Ocean, highlighting the importance of a dynamic Pacific-Southern Ocean deep-water reconfiguration for shaping late-glacial oceanic carbon storage, and subsequent deglacial oceanic-atmospheric $CO_2$ transfer.

Antarctic ice core records provide unambiguous evidence that glacial $p CO_2$ in the atmosphere was ~80 ppm lower than during the present and past interglacial periods[1,2]. Since the oceanic carbon pool is about 60 times larger than the atmosphere[3,4], the deep ocean reservoir of dissolved inorganic carbon (DIC) as well as changes in the Meridional Overturning Circulation (MOC) likely played a pivotal role in past variations of $p CO_2$. However, a consensus on the contributions of specific marine reservoirs for glacial carbon storage, their connections to major deep-water masses, and their temporal and spatial variations in storage capacity are needed to better understand glacial-interglacial $p CO_2$ change and atmosphere-ocean dynamics. The Southern Ocean, in particular the large Pacific sector, likely played an important role in deglaciation, given its size and

ability to directly connect the deep oceanic reservoirs with the atmosphere via surface exchange processes[5–10].

During the last glacial maximum (LGM), most paleoclimate proxy reconstructions indicate a generally shallower Atlantic MOC[11,12]. Overall, this slowed physical ocean movement resulted in increased nutrient and carbon concentrations in the deep ocean, particularly within the deep glacial Pacific and the Southern Ocean, resulting in the isolation of these biogeochemical constituents from the surface ocean and atmosphere during the LGM[11–15]. Both the year-round expansion of sea ice around Antarctica[16] and a northward shift of the Southern Westerly Wind Belt (SWW) is thought to be underlying forcing factors for this glacial pattern[17,18]. While direct evidence of carbon storage and release in this area is scarce, previous studies[19–24] showed that the deep

[1]Japan Agency for Marine-Earth Science and Technology, Research Institute for Global Change, Natsushima-cho 2-15, Yokosuka 237-0061, Japan. [2]MARUM-Center for Marine Environmental Sciences, University of Bremen, Leobener Straße 8, 28359 Bremen, Germany. [3]Alfred-Wegener-Institut Helmholtz-Zentrum für Polar und Meeresforschung, Am Alten Hafen 26, 27568 Bremerhaven, Germany. [4]Leibniz Institute for Baltic Sea Research, Warnemünde, 18119 Rostock, Germany. [5]The University of Tokyo, Atmospheric and Ocean Research Institute, 5-1-5, Kashiwanoha, Kashiwa 277-8564, Japan. [6]These authors contributed equally: Shinya Iwasaki, Lester Lembke-Jene, Kana Nagashima, Helge W. Arz, Naomi Harada, Katsunori Kimoto, Frank Lamy. ✉ e-mail: siwasaki@marum.de

Pacific (below ~2000 m) was relatively less ventilated during the glacial period, resulting in additional carbon accumulation in already carbon-rich, old Pacific and Southern Ocean deep waters. Previous studies hypothesized that such old deep water in the Pacific and the Southern Ocean became a source of carbon efflux in the following deglaciation. In fact, it was suggested that glacial Pacific Deep Water (PDW), exported into the Atlantic Ocean via the Drake Passage, played a positive feedback role in increased carbon storage in the South Atlantic during the LGM[25]. However, data directly allowing to quantify deep-water carbon chemistry and distribution are limited to the southwest and equatorial Pacific[26–28], both distal to the Drake Passage and the southernmost South Atlantic.

For the geological past, most paleoclimate interpretations of the MOC and water mass characteristics are based on measurements of foraminifera-based $\delta^{13}C$, radiocarbon ventilation ages, or radiogenic isotope proxies[21–23], which do not directly provide information on quantitative changes in carbonate chemistry or DIC budgets. Because variations in deep-water carbonate ion concentration ($[CO_3^{2-}]$) are primarily governed by DIC and alkalinity, the determination of quantitative $[CO_3^{2-}]$ is a means to estimate deep ocean carbon storage.

The B/Ca ratio of epifaunal benthic foraminifera, particularly *Cibicidoides wuellerstorfi*, has so far been the most-often deep-water $[CO_3^{2-}]$ proxy, developed on the basis of empirical correlation with the deep-water carbonate saturation state ($\Delta[CO_3^{2-}]$)[25,29,30]. For the Pacific sector of the Southern Ocean, B/Ca data from the subpolar Southwest Pacific show a significant decrease of ~15 $\mu$mol kg$^{-1}$ in $[CO_3^{2-}]$ within the UCDW during the LGM. This supposedly higher glacial DIC storage was followed by ~20 $\mu$mol kg$^{-1}$ increase during the early deglacial, indicating subsequent DIC release to the atmosphere[26]. However, because the B/Ca method requires mono-specific epifaunal benthic foraminifera for measurements, which multiple important Pacific and Southern Ocean locations are often lacking, $[CO_3^{2-}]$ reconstructions to date are restricted to a few sites, almost exclusively with water depth of above ~2500 m[26,27].

Deep-water $\Delta[CO_3^{2-}]$, with $[Ca^{2+}]$ and $[CO_3^{2-}]$ as variables, also largely control the dissolution of planktic foraminiferal tests, which are widely distributed in deep-sea sediments[31]. Because $[Ca^{2+}]$ in the ocean

has a long residence time (~10$^6$ years), it can be assumed to remain constant on a time scale shorter than 100 ka[32]. Therefore, deep-water $\Delta[CO_3^{2-}]$ is primarily governed by $[CO_3^{2-}]$ on glacial-interglacial time scales. In addition, the empirical correlation between carbonate dissolution and deep-water $\Delta[CO_3^{2-}]$ suggests that measurements of planktic foraminiferal test dissolution intensity can provide information about past $[CO_3^{2-}]$. The size-normalized weight (SNW) of planktic foraminiferal tests is a conventional proxy of carbonate dissolution intensity in under-saturated water[33,34]. However, the surface water carbonate chemistry significantly affects the calcification intensity of planktic foraminifera and alters their test thickness prior to carbonate dissolution after death and deposition at the seafloor, thus can affect the SNW proxy as well[35–37]. This nature of the SNW proxy poses challenges to using it as a tool to quantitatively reconstruct deep-water $[CO_3^{2-}]$ in the areas, where glacial-interglacial sea surface environmental variation was large.

Here, we employ a proxy approach to derive $[CO_3^{2-}]$ data based on the dissolution of planktic foraminiferal tests of *Globigerina bulloides* (*G. bulloides*) by using the test density analyzed by X-ray micro-computer tomography scanning (XMCT). This approach also enables us to assess interior test conditions and test wall thickness directly. Focusing on the micro-scale density distribution in specimens, this proxy is not affected by geometric characteristics of tests like size or thickness, which are influenced by sea surface conditions. Therefore, our method overcomes the weaknesses of conventional dissolution proxies[38–40], while showing a significant correlation with deep-water $\Delta[CO_3^{2-}]$[39]. We apply this method to reconstruct $[CO_3^{2-}]$ changes in the subantarctic SE Pacific from the LGM across the deglaciation (10–21 ka BP). We studied a depth transect comprising four sediment cores collected off the southern Chilean margin and in the abyssal South Pacific from water depths between 1500 and 4100 m (Fig. 1). These sites are located in a sensitive position to distinguish between Northern vs. Southern-sourced water mass contributions to the deep oceanic carbon pool prior to their export into the Atlantic via the Antarctic Circumpolar Current (ACC) through the Drake Passage. Our $[CO_3^{2-}]$ reconstructions from these sediment cores enable us to infer the vertical distribution of the deep-ocean carbon reservoir in Circumpolar Deep Water (CDW) and the influence of mixing with surrounding deep-water masses, i.e.,

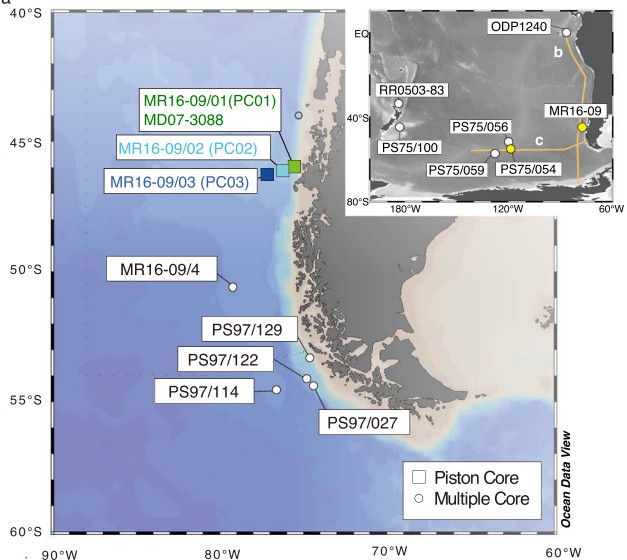

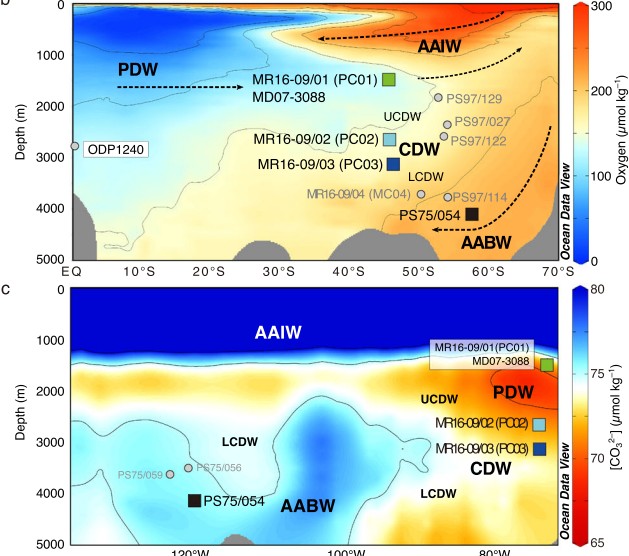

**Fig. 1 | Cores location and study area. a** Location of piston cores of MR16-09 (PC01: 46° 04′S, 75° 41′W; 1537 m water depth, PC02: 46° 04′S, 76° 32′W; 2787 m water depth, PC03: 46° 24′S, 77° 19′W; 3074 m water depth), and location of multiple cores analyzed in this study. The enlarged map shows locations of PS75/054−1 (56°S, 115°W, 4085 m) and core sites referred to in this study. **b** Cross-sections of

oxygen concentration ($\mu$mol kg$^{-1}$) along ~80°W and **c** seawater $[CO_3^{2-}]$ ($\mu$mol kg$^{-1}$) along ~47°S based on the data from Global Ocean Data Analysis Project (GLODAP). The distribution of principal water masses (PDW Pacific Deep Water, AAIW Antarctic Intermediate Water, AABW Antarctic Bottom Water, CDW Circumpolar Deep Water) is shown. Map and sections generated using Ocean Data View[69].

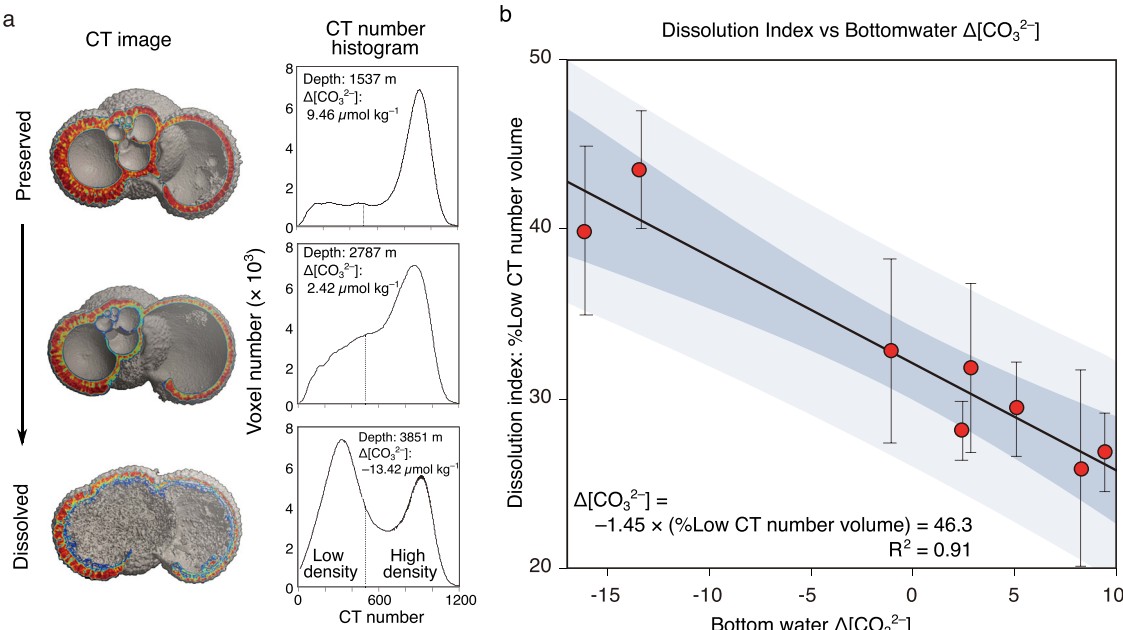

**Fig. 2 | Test dissolution process and CT-based proxy calibration with deep-water $\Delta[CO_3^{2-}]$ using core-top samples. a** Variations in cross-section iso-surface images and Computer tomography (CT) number histograms of *G. bulloides* tests with dissolution. Typical condition of tests is selected from each sample and shown. Tests were obtained from the three core-top samples from depths of 1537, 2787, and 3851 m, and deep-water $\Delta[CO_3^{2-}]$ of 9.46, 2.42, and −13.42 μmol kg⁻¹, respectively. The dashed line in the CT number histograms shows the threshold between low and high values. **b** Plots of %Low-CT-number calcite volume of *G. bulloides* tests against deep-water $\Delta[CO_3^{2-}]$ at eight sampling sites of multiple cores. The regression line and the 95% confidence interval (dark shading) and the prediction interval (light shading) are shown.

PDW and Antarctic Bottom Water (AABW) (Fig. 1). Based on this depth transect reconstruction of $[CO_3^{2-}]$, we suggest a dynamic, but transient, deglacial redistribution within the marine carbon reservoir in the different deep Southern Ocean water masses that help to elucidate dynamical carbon storage and release pathways during atmospheric $p$CO₂ changes increases.

## Results and discussion
### Application of $[CO_3^{2-}]$ proxy to the SE Pacific
A dissolution index obtained from XMCT scanning provides planktic foraminiferal test density values for the cosmopolitan species *G. bulloides*. It enables us to exclude the effects of wall thickness, which change with sea surface conditions, and therefore provides a reliable method to evaluate the dissolution intensity of this species' tests.

The internal structure of *G. bulloides* tests shows selective dissolution of inner calcite, and CT number histograms changed from mono-modal to bi-modal distribution with the progress of dissolution (Fig. 2a). This indicates that the relative volume of low-CT number calcite increases with dissolution and is applicable as a quantitative dissolution index, confirming earlier studies[39,40]. Calibration between the dissolution index, defined as %Low-CT-number calcite volume in this study, and deep-water $\Delta[CO_3^{2-}]$ at each core site in the Southeast Pacific shows that the %Low-CT-number calcite volume is effective as a quantitative proxy of deep-water $\Delta[CO_3^{2-}]$, precisely at the bottom water-sediment interface for the study region (Fig. 2b and Supplementary Data 1). Our regression equation is as follows:

$$\Delta[CO_3^{2-}] = -1.45 \times (\%Low-CT-number\ calcite\ volume) + 46.3, R^2 = 0.91 \quad (1)$$

Based on this equation, $[CO_3^{2-}]$ at each core site was calculated using the following equation:

$$[CO_3^{2-}] = -1.45 \times (\%Low-CT-number\ calcite\ volume) + [CO_3^{2-}]_{sat} \quad (2)$$

The $[CO_3^{2-}]_{sat}$ is a constant depending on the bottom water temperature, salinity, and pressure at each core site. The overall effects of these parameters in the deep ocean $[CO_3^{2-}]_{sat}$ on glacial-interglacial time scales change ~0.5 μmol kg⁻¹, based on ~3 °C change in bottom water temperature, ~1.5 psu change in salinity, and ~120 m equivalent change in pressure[41]. Therefore, modern $[CO_3^{2-}]_{sat}$ values are employed to calculate down-core $[CO_3^{2-}]$. Based on the standard error of individual % Low-CT-number calcite volume measurement in each sample and our established calibration using multiple core samples, the uncertainty associated with reconstructing deep-water $[CO_3^{2-}]$ is ~5.0 μmol kg⁻¹ at the 95% confidence level. This makes it possible to detect ~10 μmol kg⁻¹ variation of $[CO_3^{2-}]$ on millennial or longer time scales.

### High-sedimentation rate interval in the deglaciation (18–18.4 ka BP)
A short-term, but distinct carbonate preservation event lasting from 18–18.4 ka BP is evident in our bathyal cores PC02 and PC03 (Fig. 3b, c). Based on the age model for our sediment cores, a significant maximum in sediment deposition occurs, as evidenced by a 50–100-fold sedimentation rate increase to ~200 cm ka⁻¹ during this time interval (Supplementary Fig. 1c). At the sites of PC02 and PC03, the XRF-scanning-derived Ti/K ratios yield simultaneous maxima, suggesting higher input of terrestrial materials from the volcanic area of the Andes Cordillera[42] during that interval (Supplementary Fig. 1b). We thus assume that a transient high-sedimentation event of terrestrial material likely promoted carbonate preservation at the seafloor[15,43], with foraminiferal test dissolution being in temporary disequilibrium with ambient deep-water. Therefore, we consider that reconstructed $[CO_3^{2-}]$ values during this short 400-year period between 18 and 18.4 ka BP are unreliable, and do not consider these data points in the following discussion.

### Deglacial $[CO_3^{2-}]$ reconstruction: evolution of deep SE Pacific carbonate chemistry
The deep-water structure in the SE Pacific around 45°S nearby our study sites is characterized by contributions of the following principal

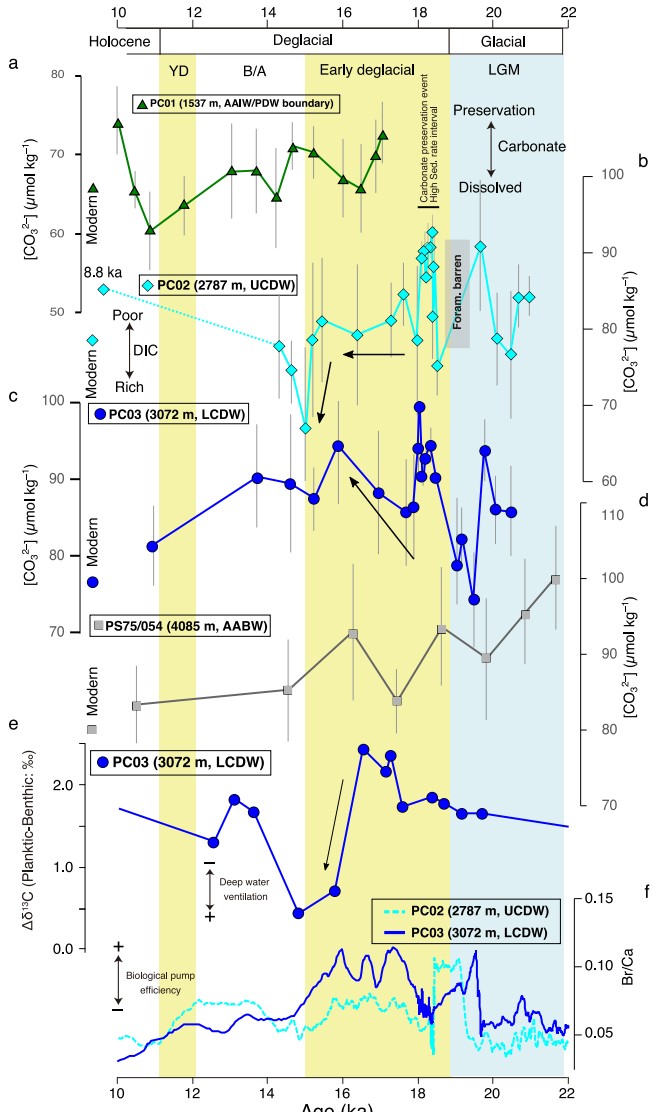

**Fig. 3 | Reconstruction of deep-water [CO₃²⁻] during 10–22 ka BP. a** PC01,
**b** PC02, **c** PC03, and **d** PS75/054. **e** Difference in stable carbon isotope ratio (Δδ¹³C)
between planktic and benthic foraminifera, and **f** Br/Ca ratio in PC02 and PC03. YD
Younger Dryas, BA Bølling-Allerød, LGM Last Glacial maximum, DIC Dissolved
inorganic carbon, UCDW Upper Circumpolar Deep Water, LCDW Lower Cir-
cumpolar Deep Water.

water masses (water mass definition based on neutral density[44]):
Antarctic Intermediate Water (AAIW: ~500–1300 m), which flows
northward, PDW (~1300–2700 m) flowing southwards, AABW
(below ~3800 m), which moves northwards. The rest is commonly
defined as CDW between ~2700–3800 m depth, leading to varying
degrees of mixing with the neighboring upper and lower water
masses. The modern CDW is divided into Upper CDW (UCDW),
relatively more influenced by PDW characteristics, and Lower CDW
(LCDW), relatively more influenced by AABW and aged modified
North Atlantic Deep Water (NADW), which is transported by the
ACC from the Atlantic[45] (Fig. 1). Among our four sediment cores
with water depths ranging from ~1500 to 4100 m (Fig. 3a–d and
Supplementary Data 2), our first and shallowest mesopelagic site
PC01 (46°04 S, 75°41 W, 1535 m) is located at the AAIW/PDW
boundary. During the early deglaciation (15–19 ka BP), site PC01
(AAIW/PDW) showed the [CO₃²⁻] ranged from 65 to 73 μmol kg⁻¹.
Thereafter, site PC01 (AAIW/PDW) values decrease to around
60 μmol kg⁻¹ at the end of the Younger Dryas (11.5 ka BP) (Fig. 3a).

Our second, upper bathyal site PC02 (46°04S, 76°32W, 2793 m) is
bathed in the uppermost part of CDW (UCDW) currently with the
influence of PDW. During the LGM (19–21 ka BP), PC02 (UCDW)
shows values between 77 and 90 μmol kg⁻¹, followed by a short,
~1000 year foraminifera-barren interval at the end of the LGM (19 ka
BP), supposedly driven by a carbonate dissolution event due to low-
[CO₃²⁻] deep water intrusion. Thereafter, the [CO₃²⁻] in PC02
(UCDW) showed stable values of ~80 μmol kg⁻¹ during the early
deglaciation (17–19 ka BP), followed by a significant decrease to
67 μmol kg⁻¹ towards ~15 ka BP, implying a transient carbon-rich
deep-water injection (Fig. 3b). Our third, lower bathyal site PC03
(46°24 S, 77°19 W, 3072 m) is bathed in the lower part of CDW
(LCDW) currently with the influence of aged NADW and AABW.
During the LGM (19–21 ka BP), PC03 (LCDW) shows values between
85 and 93 μmol kg⁻¹, while it shows lower values ~80 μmol kg⁻¹ at the
end of the LGM (~19 ka BP). Thereafter, during the early deglaciation
(15–19 ka BP), PC03 (LCDW) shows significant (Student's t test;
t = −15.2, df = 14, p < 0.01) [CO₃²⁻] increases from 75 to 95 μmol kg⁻¹,
implying a marked DIC reduction in LCDW (Fig. 3c). Finally, our
deepest, abyssal site, PS75/054-1 (56 °S, 115 °W, 4085 m), is cur-
rently bathed in AABW. In contrast to our three shallower sites, it
shows a decreasing [CO₃²⁻] trend from relatively high glacial
values of 100 μmol kg⁻¹ towards Early Holocene values of
84 μmol kg⁻¹ (Fig. 3d).

## Factors influencing [CO₃²⁻] changes: potential effects of export production and vertical mixing

Variations in the geochemical characteristics of deep-water masses are
a factor that potentially alters deep water [CO₃²⁻]. In previous studies,
ventilation age reconstruction (i.e., ¹⁴C age difference between planktic
and benthic foraminifera) suggested that South Pacific deep-water
masses, deeper than ~ 2000 m, were less ventilated and more isolated
from the atmosphere, in line with a presumed strong stratification,
during the LGM (19 ka BP), followed by enhanced mixing with well-
ventilated surface water during the early deglaciation (15–19 ka
BP)[15,22,43,46]. In addition, radiogenic isotope results, which indicate a
contribution from AABW supplied from the Ross Sea, suggested the
break up of deep bathyal water stratification after the LGM and
enhanced mixing of AABW into shallower water masses[23]. In our study,
the results of the difference in stable carbon isotope between planktic
and benthic foraminifera (Δδ¹³C planktic-benthic), which represents
the strength of bottom-surface ventilation[21], suggested that ventilation
has been poor at the LGM (19 ka BP) in bathyal PC03. Thereafter, it
shows enhanced ventilation at the end of early deglaciation (15 ka
BP) (Fig. 3e).

On the other hand, remineralization of organic matter in the
surface sediments may locally decrease porewater [CO₃²⁻] and
thus induce carbonate dissolution. On the Chilean margin, varia-
tions in the bulk sediment Br/Ca ratios can serve as an indication of
organic carbon delivery to the seafloor. This is because sedimen-
tary bromine (Br) and calcium (Ca) are closely associated with
biogenic organic carbon and carbonate content, respectively[47].
Furthermore, the carbon isotope gradient represented by Δδ¹³C
planktic-benthic is also related to biological carbon pump
efficiency[48,49]. Thus, high Δδ¹³C planktic-benthic implies effective
organic carbon transport from surface to deep ocean. In our study,
there appears to be no consistent relationship between either
proxy and carbonate preservation (Fig. 3e, f). Thus, even if dif-
ferences in productivity and/or organic carbon export existed at
our bathyal sites, it is unlikely that the organic carbon burial and
decomposition in the sediment surface was the principal control-
ling factor of carbonate dissolution intensity at our study sites.
Therefore, we consider the carbonate chemical condition in deep-
water masses at the sites of our study (i.e., AAIW/PDW, CDW, and
AABW) to be principally reflecting a true water mass signal.

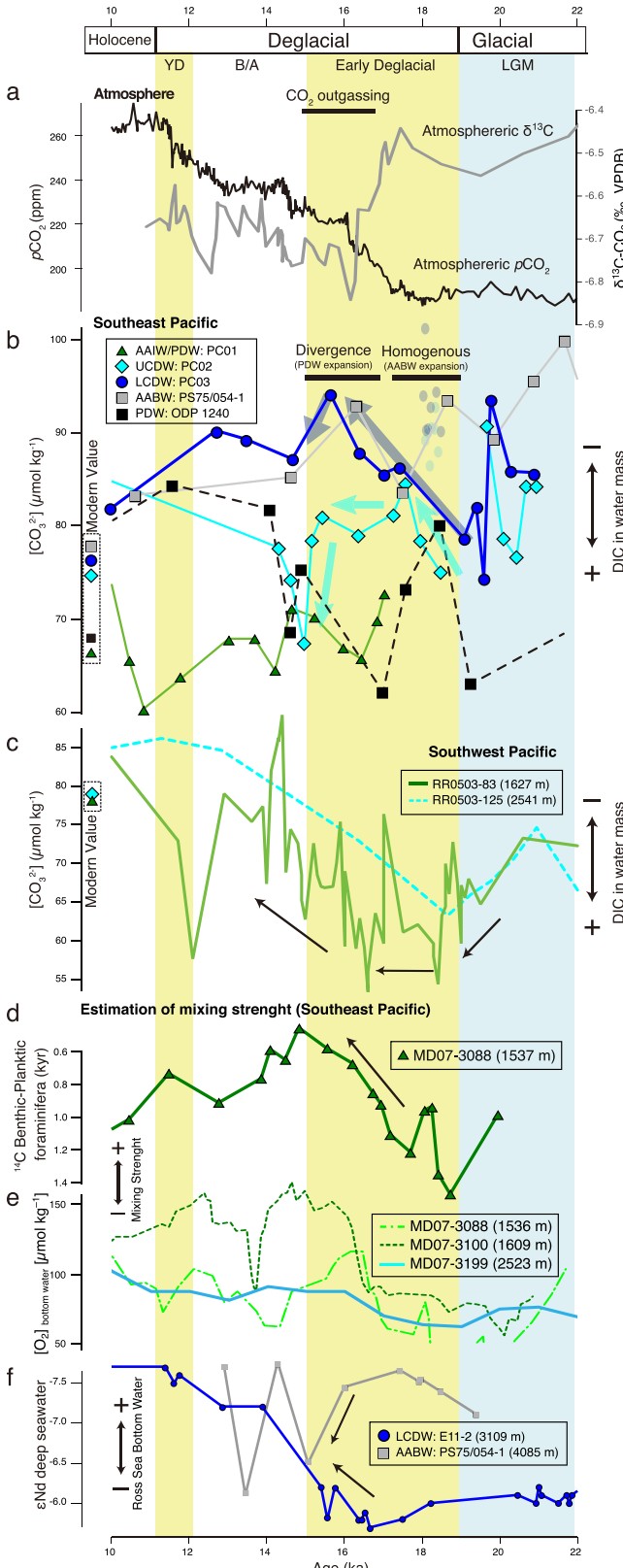

**Fig. 4 | Deep-water [CO₃²⁻] change in study area compared to the records in the South Pacific.** **a** Atmospheric CO₂ concentrations[70], and atmospheric δ¹³C record from Antarctic ice cores[71]. **b** Deep-water [CO₃²⁻] based on XMCT scanning method in the SE Pacific (this study) and result from ODP1240 from the Eastern equatorial Pacific[28]. **c** Deep-water [CO₃²⁻] based on benthic foraminiferal B/Ca proxy in the SW Pacific[26,27]. **d** Changes in the records of foraminifera benthic–planktic ¹⁴C age in the SE Pacific[21]. **e** Changes in the records of bottom water oxygen concentration reconstruction in the SE Pacific[50]. **f** Change in εNd of deep water in the Southern Pacific[23].

## Transient zonal and meridional dynamics of Pacific deep-water carbonate chemistry

In theory, changes in deep water [CO₃²⁻] at a given site can reflect either shift in deep-water circulation or changes in the ambient water DIC content. In either case, the deep-water [CO₃²⁻] variation would imply a change in carbon storage. In this study, the variations in [CO₃²⁻] at bathyal sites PC02 and PC03 fluctuate between the [CO₃²⁻] of the mesopelagic site PC01 and abyssal site PS75/054-1 (Fig. 4b), implying that the deep-water masses at bathyal sites around 2000–3000 m water depth are principally a mixing product of the surrounding water masses.

A previous reconstruction of deep-water ventilation on the Chilean margin, using radiocarbon ventilation ages, suggested a strong stratification and effect of less ventilated water on water depths of ~1500 m close to PC01 at the end of the LGM (19 ka BP)[21]. Thereafter, during the early deglaciation (15–16.5 ka BP), the water mass close to PC01 was characterized by the effect of well-ventilated water (Fig. 4d)[21]. In the SE Pacific, the deeper water mass ventilation has also been assessed based on bottom water oxygen concentration reconstruction using benthic foraminifera faunal assemblages[50]. Compared to results from a core at a shallower depth (~1600 m), the core at a depth of ~2500 m close to PC02 indicates a continuous influence of less ventilated deep water during the LGM to the early deglacial (Fig. 4e)[50]. Our deep-water [CO₃²⁻] reconstructions at Site PC02 show a decrease during the early deglaciation (15–16.5 ka BP), supporting the results of a continuous influence of less ventilated deep water (Fig. 4b). On the other hand, the [CO₃²⁻] at Site PC01 was similar to PC02 at the end of early deglaciation (15 ka BP), which appears to contradict the results of existing ventilation age reconstructions[21]. In this context, it is important to highlight that ventilation age is an indirect proxy of carbonate chemistry. In theory, the ventilation age results could be reconciled with our results if the chemical properties of the studied water mass were significantly different and its carbonate chemistry was primarily related to its ventilation age. However, the low [CO₃²⁻] reconstruction obtained for Site PC01 with our method implies that the water mass at Site PC01 was located at the boundary of AAIW and PDW. Therefore, it should have been affected by aged PDW during the early deglacial at least as strongly as at present, which is not consistent with observed large differences in ventilation ages[21].

Upstream of our core sites, the [CO₃²⁻] of PDW was reconstructed in the Equatorial East Pacific at Site ODP1240 (2921 m: shown in Fig. 4b), using the B/Ca ratio of benthic foraminifera, suggesting aged, low [CO₃²⁻] PDW during the LGM[28]. On the other hand, our deepest, abyssal site PS75/054-1, which showed a continuous decrease in the [CO₃²⁻] after the LGM, implies a strong effect of AABW supplied from the Ross Sea during the LGM. Thereafter, previous studies suggested mixing with bathyal water masses from above during the early deglaciation based on εNd analyses[23]. Therefore, we assume that the [CO₃²⁻] of CDW as recorded in our bathyal sites PC02 and PC03 is mainly controlled by varying mixing of northern-sourced low-[CO₃²⁻] PDW and southern-source higher-[CO₃²⁻] AABW.

At the end of the LGM (19 ka BP), our bathyal [CO₃²⁻] reconstructions from sites PC03 differ significantly from values at ODP1240 (PDW) and PS75/054-1 (AABW). Considering that the South Pacific deep-water mass structure was supposedly strongly stratified during this period (Fig. 4d, f)[21,23], glacial CDW should have been less influenced by the overlying PDW supplied from north and the underlying AABW supplied from south. A glacial CDW isolated from surrounding water masses was also inferred for the Atlantic Southern Ocean, and explained by a shallower depth and a higher flux of glacial intermediate water[20,51]. Specifically, reconstructions in the subantarctic Southern Atlantic revealed that the Atlantic CDW at the depth of ~3800 m showed lower [CO₃²⁻] (~70 μmol kg⁻¹) than underlying Atlantic AABW (~90 μmol kg⁻¹) at the depth of ~5000 m[25] (Fig. 5b), which suggest the isolation of glacial Atlantic CDW from the other water masses. Such

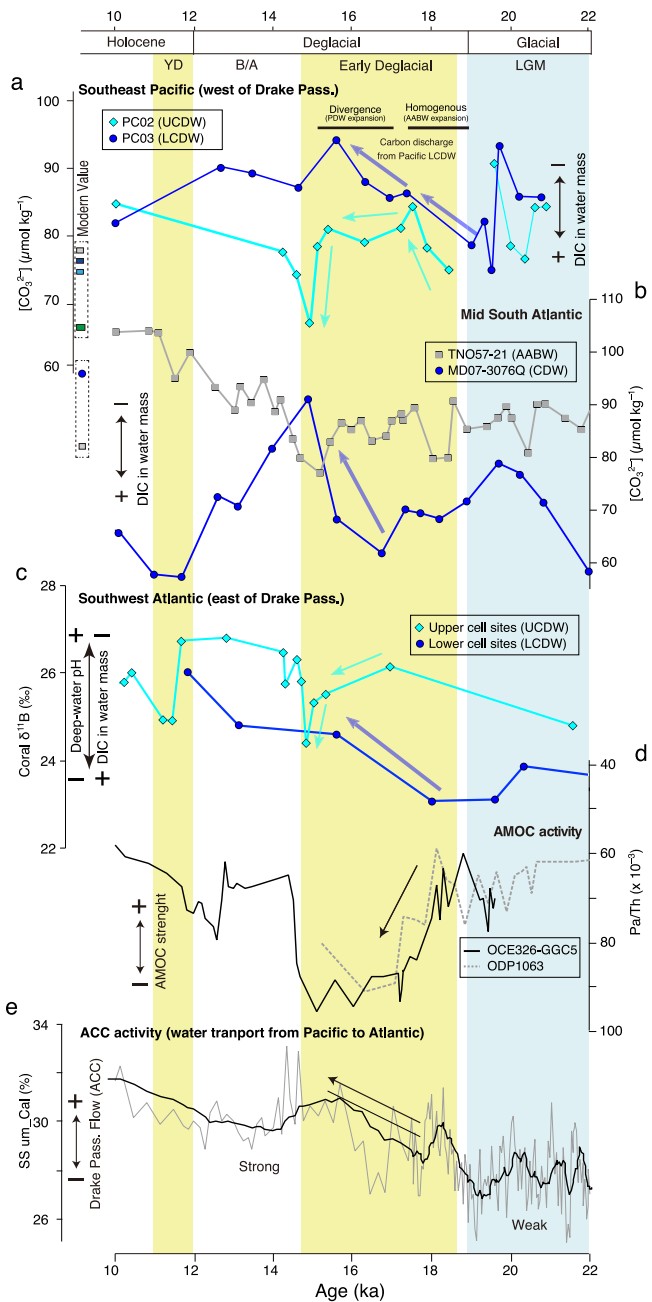

**Fig. 5 | Deep-water [CO$_3^{2-}$] change in study area compared to the records in the South Atlantic across the Drake Passage. a** Deep-water [CO$_3^{2-}$] reconstruction by XMCT scanning method in the SE Pacific (this study). **b** Deep-water [CO$_3^{2-}$] reconstruction by benthic foraminiferal B/Ca proxy in the Southern Atlantic[25]. **c** Deep-water pH reconstruction by deep-sea coral δ$^{11}$B[56]. **d** Variance of the Atlantic meridional overturning circulation (AMOC) strength shown by $^{231}$Pa/$^{230}$Th[72,73]. **e** Variance of grain size in the sediment core at the Drake Passage, represents the strength of deep-water flow[55].

Atlantic CDW, which partly originated from glacial NADW, may have been channeled into the South Pacific by the ACC. Furthermore, glacial Pacific CDW may have been re-supplied into Atlantic CDW via the Drake Passage.

We now discuss the varying contributions of surrounding water masses to CDW during the deglaciation (15–19 ka BP). During this period, the Atlantic MOC was weakened (Fig. 5d), with suppressed NADW supply. Furthermore, the deep-water stratification in the Southern Pacific was weakening[21–23,50] (Fig. 4d–f), such that the CDW could then become controlled by both the inflow of northern-sourced

shallower PDW and mixing with the southern-sourced deeper AABW. At the beginning of the early deglaciation (18 ka BP), the bathyal [CO$_3^{2-}$] in both PC02 and PC03 were ~85 μmol kg$^{-1}$, converging to similar values with that of abyssal site PS75/054-1, suggesting that CDW was temporally homogenized with a more pronounced AABW. At that time, SE Pacific deep-water masses were still isolated from the atmosphere, as evidenced by ventilation age reconstruction in nearby core MD07-3088[21] and Δδ$^{13}$C (planktic -benthic foraminifera) in PC03 (Fig. 3e). Thereafter, the [CO$_3^{2-}$] in our bathyal sites PC02 and PC03 gradually diverged, with the shallower core PC02 converging to northern-sourced PDW signatures towards the end of early deglaciation (15 ka BP). This implies that the influence of northern-sourced PDW gradually increased and reached a maximum expansion at the end of the early deglaciation (15 ka BP).

Our quantitative reconstruction of SE Pacific deep-water [CO$_3^{2-}$] content enables us to compare our location to other key regions and to assess potential connections and spatial distributions of deep-water carbon storage. As a first step, we compare the variations in deep-water carbon storage based on deep-water [CO$_3^{2-}$] between the SW Pacific (Fig. 4c)[26,27], and the SE Pacific (this study; Fig. 4b). Under modern conditions, the main flow of abyssal AABW to the north is located in the SW Pacific within the Deep Western Boundary Current system. In contrast, the return flow of DIC-rich and low-[CO$_3^{2-}$] PDW is mostly located in the SE Pacific at bathyal depths of 2000–3500 m[52]. In the subantarctic SW Pacific, the variations in [CO$_3^{2-}$] were recently reconstructed along a transect located off New Zealand at three sites from 1160 m, 1630 m, and 2540 m water depth, respectively, by using B/Ca ratios of benthic foraminifera[26,27]. In particular, the mesopelagic SW Pacific showed low and stable [CO$_3^{2-}$] values of ~60 μmol kg$^{-1}$ at the beginning of the early deglaciation (17–19 ka BP), followed by a mid-deglacial 15–20 μmol kg$^{-1}$ increase beginning around 17 ka BP (Fig. 4c). These results provide evidence for a reduction in deep-water carbon storage accompanied by strengthened vertical mixing at the end of the early deglaciation[22]. Compared to our vertical profile of [CO$_3^{2-}$] in the SE Pacific without [CO$_3^{2-}$] increase in mesopelagic and upper bathyal sites, this suggests that a reduction in deep-water carbon storage occurred in shallower depths in the SW rather than in the SE Pacific (Fig. 4b, c). This asymmetrical east-west imbalance in Southern Pacific carbon storage was most likely caused by the preferential low-[CO$_3^{2-}$] PDW import from the north into the SE Pacific. The ventilation age reconstructions in the Pacific Southern Ocean during the early deglaciation also revealed that the rejuvenation (i.e., reduction in ventilation ages) of deep-water mass was more pronounced in the SW Pacific (from 3 to 1 kyr) than in the SE Pacific (from 1.5 to 1 kyr)[21,22], i.e., ventilation of the deep-water was more significantly enhanced in the SW Pacific during the early deglaciation. This also supports the presumed east-west imbalance in South Pacific [CO$_3^{2-}$] variations (Fig. 4). Thus, spatial reconstructions of deep-water masses and their biogeochemical signatures in both meridional (north-south) and zonal (east-west) directions are indispensable for a quantitative and process-oriented understanding of variations in marine carbon storage and its transfer between different reservoirs.

## Pacific-Atlantic deep-water export through ACC transport: Implications for the Southern Ocean carbon budget

Our cores from the SE Pacific Chilean margin, close to the Drake Passage, enable us to better assess the transport and carbonate chemistry of deep-water masses exchanged between the Pacific and Atlantic. The intensity of Drake Passage throughflow is governed by the ACC, and past changes in throughflow over the last glacial have been reconstructed by grain size and geochemical analyses[53–55]. These results suggest that the Drake Passage throughflow was weaker during the LGM and gradually strengthened across the termination, with a maximum at the end of the early deglaciation (Fig. 5e). On the downstream side of Drake Passage in the western South Atlantic, deep-water pH

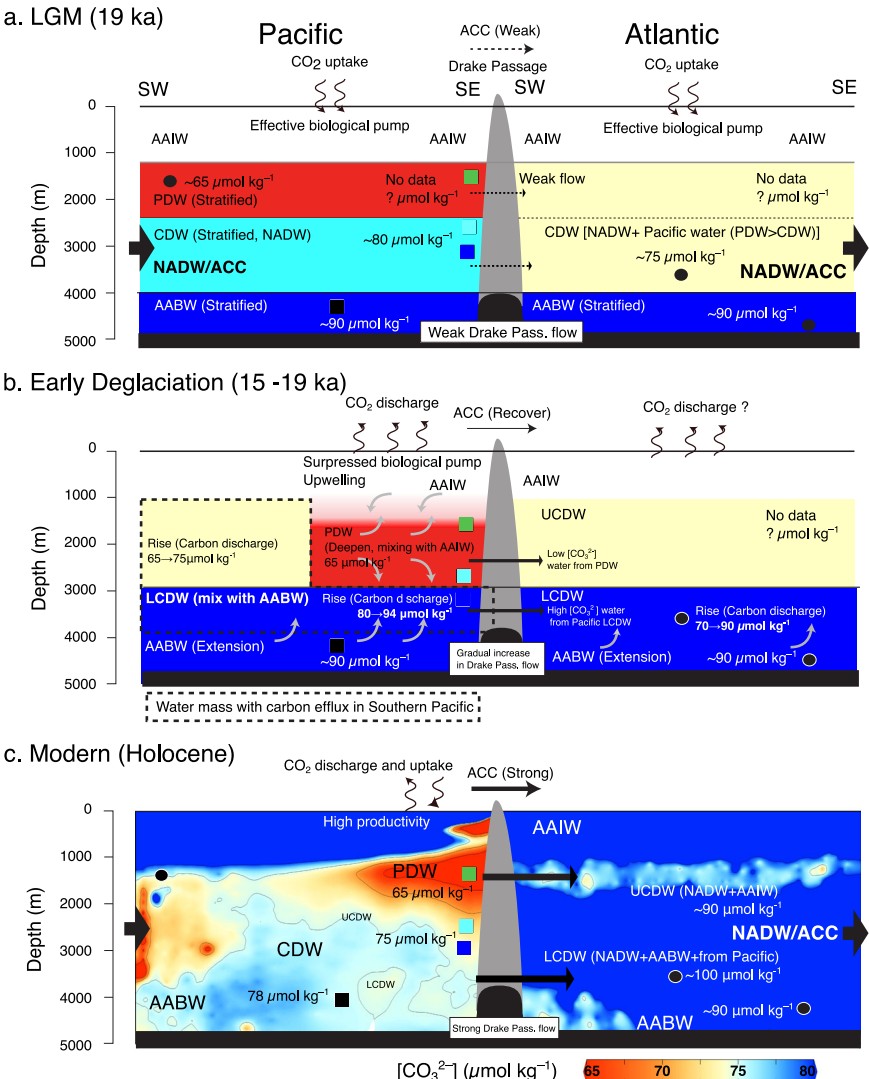

**Fig. 6 | Schematic illustration of water column structure in the Southern Pacific and the Southern Atlantic at the end of LGM, the early deglacial, and Modern condition. a** LGM (Last Glacial Maximum, 19 ka BP): characterized by strong deep-water stratification, weak Antarctic Circumpolar Current (ACC), weak Drake Passage flow, and strong Atlantic meridional overturning circulation (AMOC). **b** The early

deglaciation (15–19 ka BP): break up of deep-water stratification, strengthening of ACC, strengthening of Drake Passage flow, and weakening or stop of AMOC. **c** Modern condition: moderate mixing of deep-water, Strong ACC, Strong Drake Passage flow, and strong AMOC. West-East transect of $[CO_3^{2-}]$ at -30°S is based on the modern data from GLODAP. Section generated using Ocean Data View[69].

after the LGM was reconstructed using deep-sea coral boron isotope data (Fig. 5c)[56,57]. These studies suggest that the deep-water pH significantly decreased at 15 ka BP at the sites within the upper over-turning cell affected by Atlantic UCDW. On the other hand, a continuous increase during 15–18 ka BP was shown in the sites within the lower overturning cell affected by Atlantic LCDW[56,57]. These variations are consistent with $[CO_3^{2-}]$ variations in the Pacific UCDW (PC02) and LCDW (PC03) at the western, upstream side of the Drake Passage shown in our study. This suggests that the deep-water transport from the Pacific to the Atlantic contributed to the carbon redistribution during the early deglaciation and was enhanced with the strengthening of Drake Passage through flow[53–55]. In addition, in the central and eastern South Atlantic, variations in $[CO_3^{2-}]$ of CDW and AABW were reconstructed with B/Ca ratios of benthic foraminifera[25], showing different values between CDW and AABW during the LGM and early deglaciation (16.5–19 ka BP), while the $[CO_3^{2-}]$ of CDW increased after 16.5 ka and showed similar values as AABW at 15 ka BP (Fig. 5b). Based on the data from the Pacific and Atlantic sectors of the Southern Ocean, we provide evidence for a temporal change in the Southern Ocean's deep-water structure, with a more pronounced influence of

deep-water transport from the Pacific to the Atlantic on the carbon cycle after the LGM (Fig. 6).

During the LGM (Fig. 6a), surface biological productivity in the subantarctic South Pacific is thought to have generally been high due to iron-rich conditions compared to interglacial periods[58,59]. Existing data from the Chilean margin suggest that the biogenic productivity and biological carbon pump efficiency were higher during the LGM than during the deglaciation[47,53], which contributed to an effective carbon uptake from the atmosphere. Considering the observation that the glacial deep Southern Ocean has been more stratified in the bathyal to the abyssal domain than today[22,23], Pacific CDW was likely more isolated from the surrounding water masses, ultimately leading to DIC-rich, aged PDW flowing eastward through the Drake Passage. Our $[CO_3^{2-}]$ reconstruction implies that carbon storage was larger in bathyal Pacific CDW than in AABW (Fig. 5a), and therefore, Atlantic southern-sourced deep-waters were likely affected by such carbon-rich PDW-influenced CDW export from the SE Pacific via the Drake Passage.

During the early deglaciation (Fig. 6b), the temporal variation of deep-water masses in the Southern Ocean changed in two steps. Firstly, at the beginning of the early deglaciation (17–19 ka BP), our

results suggest an expansion and mixing of AABW into overlying CDW, shown by homogenous [$CO_3^{2-}$] in Pacific CDW. Deep water in the SE Pacific was still quite isolated from the surface ocean, and carbon discharge to the atmosphere was hence suppressed[21]. In addition, during this period, the AMOC weakened while the ACC transport strengthened, which likely enhanced deep-water transport from the Pacific to the Atlantic Ocean via the Drake Passage. In the central and eastern South Atlantic, on the other hand, the deep-water [$CO_3^{2-}$] reconstructions from CDW and AABW water depths yield diverging values[25], suggesting that Atlantic CDW was still relatively isolated from Atlantic AABW, although their Pacific equivalents were mixed in the South Pacific.

Secondly, at the end of the early deglaciation (15–16.5 ka BP) (Fig. 6b), the biological pump efficiency was suppressed due to an increasing effect of the carbonate counter pump[47], and enhanced upwelling resulting in carbon dioxide outgassing to the atmosphere. The deep-water export from the Pacific to the Atlantic through the Drake Passage was further strengthened, enabling a leakage of low [$CO_3^{2-}$], carbon-rich Pacific UCDW to the South Atlantic. On the other hand, deep-water [$CO_3^{2-}$] in Atlantic CDW increased and showed values similar to AABW (90 $\mu$mol kg$^{-1}$). This was seemingly caused by enhanced transport of Pacific LCDW into the South Atlantic. This supposed export of low [$CO_3^{2-}$] UCDW and high [$CO_3^{2-}$] LCDW from the Pacific to the Atlantic sector of the Southern Ocean is supported by the reconstructed variations in deep-water pH within the upper and lower overturning cells at the eastern, i.e., Atlantic side of Drake Passage representing UCDW and LCDW, respectively[57]. In addition, young and carbon-poor waters originating from upwelled northern-sourced PDW were transported into Atlantic CDW, which may have contributed to the increase in [$CO_3^{2-}$] content in Atlantic CDW. This suggests that better-ventilated (i.e., low DIC) deep water was exported to the Atlantic and mixed into Atlantic LCDW and AABW, contributing to the reduction of carbon storage in the CDW as a whole during the early deglaciation (15–19 ka BP), (Fig. 6b).

Our quantitative reconstruction of deep-water [$CO_3^{2-}$] based on the depth transect in the SE Pacific, a central junction point connecting the Pacific and the Atlantic Ocean, suggests that dynamic, transient variations in deep-water structure and carbon chemistry characteristics largely shape and determine the amount of carbon stored in the Southern Ocean. In particular, the volumetric expansion of Pacific CDW with high [$CO_3^{2-}$], caused by the deglacial reconfiguration of abyssal to bathyal waters in the Southern Ocean, likely contributed to the reduction of the carbon storage in LCDW and was followed by effective carbon discharge due to enhanced surface ocean ventilation of deep-water masses.

Our [$CO_3^{2-}$] reconstruction also enables us to roughly estimate changes in the bathyal to abyssal carbon budget after the LGM. During the LGM to early deglaciation (15–19 ka BP), the [$CO_3^{2-}$] at our bathyal site PC03 shows a 20 ± 14 $\mu$mol kg$^{-1}$ increase in the SE Pacific, implying the reduction of DIC in the water mass of LCDW between the PDW and AABW. To calculate DIC variations after the LGM using [$CO_3^{2-}$] data, additional parameters of the deep-water carbonate system (temperature, salinity, and total alkalinity) are required and need to be estimated. Both proxy and modeling studies imply that mean global deep-ocean temperature and salinity during the LGM were ~2–3 °C lower and ~1–3 PSU (practical salinity units) higher than during the Holocene[41,60,61]. While proxy-based estimates of past deep-water total alkalinity are quite uncertain, modeling studies provide evidence for glacial-interglacial total alkalinity variations and deep-water alkalinity being higher in the LGM than in the Holocene[62–64]. These model-based variations in glacial-interglacial deep-water alkalinity are ~120 $\mu$mol kg$^{-1}$ in box model simulations[62], ~25 $\mu$mol kg$^{-1}$ in proxy-based reconstructions[61], and ~10–50 $\mu$mol kg$^{-1}$ in budget estimates that considered the effects of alkalinity burial and weathering[64]. Based on the above, we assumed that the deep-water alkalinity during the LGM is

0–125 $\mu$mol kg$^{-1}$ higher than modern values so that we avoid ascribing weight to any particular deep-water alkalinity value and thoroughly explore the likely range of past values, given the available model and proxy-based constraints. As for the water parameters at the end of the early deglaciation (15 ka BP), temperature, salinity, and alkalinity ranged between the values of modern and LGM conditions. The DIC estimates based on the above assumptions suggest that a 20 ± 14 $\mu$mol kg$^{-1}$ increase in deep-water [$CO_3^{2-}$] content corresponds to ~35 − 200 ± 26 $\mu$mol kg$^{-1}$ reduction in DIC. During this period, atmospheric $p$CO$_2$ increased from 180 to 210 ppm, corresponding to a carbon efflux of ~63 GtC to the atmosphere. If deep-water DIC of LCDW in the South Pacific (assumed water mass distribution: 30°S–65°S, 75°W–180°W, 3000–4000 m water depth, with reference to the modern water mass distribution) uniformly was reduced by ~35 ± 26 $\mu$mol kg$^{-1}$ as a minimum estimated value, the amount of carbon efflux from this water mass can be calculated as ~9.4 ± 7.5 GtC. This amount would correspond to ~15 ± 11% of the total carbon emission to the atmosphere. Although our DIC and carbon efflux calculations broadly vary with the assumed values of deep ocean alkalinity and volume of target water mass, we suggest that carbon storage in South Pacific LCDW played a critical role in the atmospheric $p$CO$_2$ rise during the early deglaciation.

## Methods
### Core sampling
Three piston cores were sampled during cruise MR16-09_leg.2 of R/V Mirai (Jan. 20-Feb. 5) conducted by the Japan Agency for Marine-Earth Science and Technology (JAMSTEC) in 2017. In addition, we used piston core PS75/054-1 retrieved during cruise PS75 of R/V Polarstern. Core sites were located on the slope of Chilean margin between 1535 m to 3072 m (PC01: 46° 04′ S, 75° 41′ W, 1535 m; PC02: 46° 04′ S, 76° 32′ W, 2793 m; PC03: 46° 24′ S, 77° 19′ W, 3072 m) and in the abyssal Southeast Pacific sector of the Southern Ocean (PS75/054-1: 56° S, 115° W, 4085 m) (Fig. 1). The recovered sediment cores of PC01 and PC02 consist mainly of olive gray silty clay with intercalations of silty layers, and PC03 consists mainly of gray clay with intercalations of silty/sandy layers. Sediment cores were continuously sliced every 2.2 cm interval throughout the core. For the analysis of planktic foraminiferal test dissolution, we used 13 samples from the upper 5.31 m of PC01, 16 samples from the upper 4.32 m of PC02, and 14 samples from the upper 2.58 m of PC03, respectively. From the core of PS75/054-1, we took 1 cm thick samples every 20 cm interval and obtained ten samples in total. From each sediment sample, complete tests of planktic foraminifera (*G. bulloides*) were hand-picked from the 250–355 $\mu$m size fraction under a stereo microscope. Those tests were still intact, not fragmented nor containing fractures or holes, or peeling of the surface. Only specimens that were not obviously filled with sediments or overgrown by pyrite or other secondary minerals were chosen. Thereafter, test dissolution intensities were measured by XMCT scanning. To eliminate the influence of variance in the initial histogram setting between individuals, we excluded all tests with a thin chamber wall with a mean thickness of <10 $\mu$m (Supplementary Fig. 3b). We used eight or more tests for dissolution intensity measurements.

For calibration, we used seafloor sediment surface samples, obtaining planktic foraminiferal tests from the undisturbed uppermost tops (0–1 cm) of Multi-Corer (MUC) profiles. Three Multiple Cores and one Pilot Core were collected during cruise MR16-09 Leg 2, and four Multiple Cores were collected during cruise PS97 of R/V Polarstern in 2016. The water depths of core sites were ranging from 1537–3851 m, and deep-water $\Delta$[$CO_3^{2-}$] at core sites ranged from −16.11 to 9.46 $\mu$mol kg$^{-1}$. We calculated these data from CTD and Niskin bottle sampling and subsequent shipboard analyses during cruise MR16-09 Leg 2, as well as nearby Global Ocean Data Analysis Project (GLODAP) sites[65] (Supplementary Table 1). Eight or more hand-picked tests of *G. bulloides* from each core-top sample (250–355 $\mu$m size fraction) were

selected for measurements of the tests' SNW (µg) and scanned by XMCT.

## Age model

The age model for MR16-09 PC01 (whole core) is based on a correlation to a CALYPSO piston core MD07-3088 from the same location, with PC01[42] using Ca relative concentration records derived from high-resolution X-ray fluorescence (XRF) core scanning (Supplementary Fig. 1a and Supplementary Data 4). The age model of MD07-3088 was based on the identification of five tephra layers and 24 AMS [14]C ages of planktic foraminifera *G. bulloides*[21]. Based on the lithostratigraphic correlation, the age model of MD07-3088 was transferred to the age model of PC01. The age model for PC02 (upper 394 cm core depth) is based on the 12 AMS [14]C ages of planktic foraminifera (mixed), and this age model was transferred to the age model of PC03 (upper 230 cm core depth) based on the peak matching of Zr/Rb ratio records (Supplementary Fig. 1b and Supplementary Data 4). The age model of PC03 is also supported by 3 AMS [14]C ages of planktic foraminifera. The measurements of AMS [14]C ages of PC02 were carried out at the AMS Laboratory of the University Museum, the University Tokyo in Tokyo, Japan[66] (Supplementary Data 5). Planktic [14]C ages were calibrated to calendar ages, using the calibration software Calib 8.20 with the MARINE20 calibration curve. All ages were calibrated with reservoir ages of sea surface water established in the core MD07-3088[21]. The age model of core PS75/054-1 is based on tying to core PS75/065-1 by using XRF rubidium scans, which locates close to the core PS75/054-1 and its age model is based on the tuning of foraminiferal δ[18]O records[23].

Based on the acquired age models, we found a significantly maximum sedimentation rate layer in PC02 (Depth in core: 200–400 cm) and PC03 (Depth in core: 100–200 cm) at 18.0–18.4 ka BP, which was likely caused by an input of allochthonous sediment due to sudden discharge from the hinterland caused by Patagonian ice sheet dynamics or a turbidite deposit[67].

## X-ray micro-CT scanning

The XMCT system (ScanXmate-D160TSS105/11000, Comscantecno Co., Ltd., Kanagawa, Japan) at the Japan Agency for Marine-Earth Science and Technology, Yokosuka, Japan, was used. For 3-D observation of the foraminiferal tests, we used a high-resolution setting (X-ray focus spot diameter, 0.8 µm; X-ray tube voltage, 80 kV; detector array size, 2000 × 1336; 1500 projections/360°; 0.5 s/projection). After XMCT scanning, ConeCTexpress software (Comscantecno Co., Ltd.) was used to correct and reconstruct the tomography data. Image cross-sections were reconstructed from filtered back projections following the general principle of Feldkamp cone beam reconstruction. The CT number, indicating calcite density, was calculated based on the X-ray attenuation coefficient of each sample. In this study, a calcite standard crystal (a particle of NBS-19) was used to standardize the CT number of each test sample: the mean CT number of air and standard crystal were defined to be 0 and 1000, respectively, and the CT numbers of foraminiferal test samples were calculated according to the following equations:

$$\text{CT number} = (\mu_{\text{sample}} - \mu_{\text{air}})/(\mu_{\text{calciteSTD}} - \mu_{\text{air}}) \times 1000 \quad (3)$$

where $\mu_{\text{sample}}$, $\mu_{\text{calcite STD}}$, and $\mu_{\text{air}}$ are the X-ray attenuation coefficients of the sample, calcite standard crystal, and air, respectively.

We used Molcer Plus 3-D imaging software (WhiteRabbit Corp., Tokyo, Japan) to obtain iso-surface images and to measure mean CT numbers and CT number histograms of foraminiferal tests based on the 3-D tomography data. The mass of voxels with a size of 0.8 µm, which have a specific CT number, was calculated on the basis of iso-surface images of foraminiferal tests. The volumes of foraminiferal tests were calculated based on the total number of voxels in each

scanned test. We also measured the test wall thickness of the four chambers in the final whorl of *G. bulloides* using iso-surface images. We measured the wall thickness at randomly selected five points in each chamber wall and showed the average thickness at a total of twenty points as the mean wall thickness.

## Test dissolution index

For the estimation of carbonate dissolution intensities, we employed variations of CT number histograms obtained by XMCT scanning. The previous study[40] found that CT number histograms of dissolved *G. bulloides* test show bimodality of the dissolved and preserved calcite. Based on this result, they proposed the relative volume of low-CT-number calcite to the volume of calcite in the whole shell (%Low-CT-number calcite volume) as a more quantitative carbonate dissolution proxy for dissolution than the conventional test weight proxy. Based on this concept, we calculated %Low-CT-number calcite volume to measure the dissolution intensity of *G. bulloides* tests using the following equations:

$$\%\text{Low} - \text{CT} - \text{number calcite volume} = (V_{\text{low}-\text{CT}-\text{number calcite}}/V_{\text{whole shell}}) \times 100 \quad (4)$$

where $V_{\text{low-CT-number calcite}}$ indicates the volume of low-CT-number calcite in an individual test, and $V_{\text{whole shell}}$ indicates the volume of the whole individual test. In this study, based on the results of XMCT scanning, we classified low-CT-number and high-CT-number calcite as calcite with CT number values of 200–500 and >500, respectively.

## Sample size of specimen from sediment sample

We also assessed the necessary minimum sample size required to evaluate the test dissolution intensity from sediment samples. For this study, we scanned 30 *G. bulloides* tests obtained from two Multiple corer samples with different carbonate dissolution intensities (PS97/114, 129). In addition, we evaluated individual variability in the sediment samples, and the assessment of transitions in analytical accuracy along with the sample size (N) of tests (Supplementary Fig. 2). The analysis based on the random selection sampling suggested that the sample size (N) of 8 individuals or more can provide data with similar variation and accuracy (i.e., within the 95% confidence interval) as with a sample size of 30 or more individuals. Furthermore, the two-tailed student t-test between sample size of eight individuals and 30 individuals suggested no distinct difference between each population ($t = 0.64$, $df = 36$, $p > 0.05$). While in principle, larger sample size can provide more accurate data, we settled on scanning eight specimens for each sediment sample, as an optimal compromise considering the time and accuracy required for measurements. As a result, the standard error of calculated %Low-CT-number calcite volume would be ~±1.5%. This would be <±3.0–4.0% at the 95% confidence interval, which is necessary to detect glacial-interglacial variations on suborbital to millennial time scales.

## SNW measurement

The SNW, or more precisely measurement-based weight[68], of the planktic foraminiferal test has been considered as an indicator of test wall thickness and density, which enables us to evaluate the test calcification and/or dissolution intensity. The mean weights (µg) of 16 non-fragmented *G. bulloides* tests picked from seafloor sediment samples, sizes ranging from 250 to 355 µm, were measured with an ultra-microbalance (Cahn C-35, Thermo Electron Corp., Round Rock, TX, USA). The analytical precision of the weight measurement was ±0.3 µg (±1σ) based on 15 repeated measurements. After the measurements of shell weights, test size, i.e., the longer axis (µm) of individual tests, was measured with image analysis software (Motic Image Plus 2.1 S, Shimadzu Rika Corp., Tokyo, Japan). The SNW of *G. bulloides*

tests were calculated using the following equation based on a previous study[37,68].

$$SNW\,(\mu g) = \text{mean test weight}\,(\mu g) \times \text{mean test size}\,(\mu m : \text{each sample})/\\ \text{mean test size}\,(\mu m : \text{whole sample})$$

(5)

Based on the analyses of planktic foraminiferal tests from undisturbed sediment surface samples, the SNW of *G. bulloides* tests shows a positive correlation with the mean wall thickness of the outermost chamber (Supplementary Fig. 3a and Supplementary Data 1), indicating that the conventional proxy of SNW is governed by test wall thickness in this study area. Furthermore, the geographical distribution of test area density differs with latitude, showing low SNW (thin test wall) in lower latitudes and high SNW (thick test wall) in higher latitudes (Supplementary Fig. 3a). This is supposed to be caused by the inclusion of thin-walled individuals in the sites from lower latitudes. It remains unsure whether such thin test walls are caused by variations in sea surface conditions or differences in genotypes. However, the observed correlations between SNW, test wall thickness, and latitudes of collected samples indicate that the SNW of *G. bulloides* test is affected by multiple factors other than dissolution at the deep seafloor. Therefore, we suggest that the conventional proxy of SNW based on the bulk test weight is not an ideally suitable method for quantitative reconstruction of deep-water $[CO_3^{2-}]$ in our study region.

## Stable isotopes

Stable oxygen and carbon isotope ratios ($\delta^{18}O$ and $\delta^{13}C$; ‰ PDB) of planktic and benthic foraminifera were obtained from core MR16-09 PC03 at the Alfred-Wegener-Institute in Bremerhaven, Germany (Supplementary Data 3). From the core PC03, 27 samples after the last glacial period were selected, and five tests of *G. bulloides* (planktic foraminifera) were picked within the 250–315 $\mu m$ size range. In addition, 19 samples were selected, and up to three tests of *Uvigerina* spp. (benthic foraminifera) were picked up within the 250–500 $\mu m$ size range. The analysis of oxygen and carbon stable isotope was performed with a Thermo Finnigan MAT253 mass spectrometer connected to a Kiel IV CARBO unit. All values are reported as ‰ vs. V-PDB. Calibration was done with NIST 19 and an internal carbonate standard of Solnhofen limestone; long-term reproducibility over a 1-year period is better than 0.08 ‰ for d$^{18}$O and 0.04 for d$^{13}$C. Based on the results of duplicate analyses, the uncertainty of isotope analyses, taking into account the effect of sample selection, is less than 0.15 ‰.

## X-ray fluorescence core scanning

The sediment core samples obtained in this study were scanned for elemental analysis with an ITRAX micro-XRF scanner at Kochi Core Center, Japan (Supplementary Data 4). These results of elemental composition in each core sample were used for the age model reconstructions. XRF spectra were measured every 0.5 cm (exposure time: 15 s) with an X-ray beam generated with a 3 kW Mo target (run at 30 kV and 55 mA).

## Data availability

All relevant data generated in this study are provided in the Supplementary Information/Supplementary Data files.

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

## Acknowledgements

This study used samples and data that were collected during cruise MR16-09_leg.2 under the Joint Research Program between JAMSTEC and COPS Sur-Austral, the University of Concepción. We also used samples that were collected during the cruise PS75. We thank Prof. Carina B. Lange and the crews of the R/V Mirai and Polarstern. This study was supported by the Japan Agency for Marine-Earth Science and Technology, Japan Society for the Promotion of Science Fellowship Grant 25-5427 and 17J09017 to S.I., Grants-In-Aid for Scientific Research (KAKENHI) Grant Numbers 15H05712 to N.H., 16H04961 and 24540505 to K.K., and 18H03370 to K.N., Alfred Wegener Institute, and the Cluster of Excellence, The Ocean Floor—Earth's Uncharted Interface, funded by the German Research Foundation (DFG).

## Author contributions

S.I., F.L., and L.L. analyzed samples and wrote the main manuscript text, and made the figures. K.N., H.W.A., N.H., and K.K. also analyzed samples and reviewed this manuscript.

## Funding

## Competing interests

The authors declare no competing interests.
