## [Peer Review File · Nature Communications]

Evidence for late glacial oceanic carbon redistribution and discharge from the Pacific Southern OceanREVIEWER COMMENTS

Reviewer #1 (Remarks to the Author):

The submitted manuscript builds nicely on a body of paleoclimate research that attempts to answer the important question of the source and process for CO₂ outgassing associated with the glacial-interglacial transition. The authors develop a new proxy tool that is uniquely suited to investigate deep water carbonate chemistry dynamics and apply this tool to a transect of cores in the Pacific Southern Ocean, a region that is likely a critical zone for ultimate ocean-atmospheric CO₂ release processes. The authors present compelling results and extensively relate their results to other studies within the region, which strengthen the interpretation of their results. Most excitingly, the authors (using some basic assumptions) are able to quantitatively estimate the fraction of CO₂ (in GtC) released from the Pacific Southern Ocean. This exciting work brings together decades of research on this particular topic and region and blends multi-proxy approaches to understand the dynamics and physical processes at play. The results of this work are well suited for Nature Communications.

Specific section and line by line comments are below.

Overall: Revisions are needed for clarity, distill the important information, include important process information when discussing proxy results and eliminate abbreviations where possible. Stylistically, review to eliminate any unnecessary words to overall work towards shortening your sentences, breaking long sentences in two when needed. This will make the manuscript much easier to read and follow. Basic typos still exist within this final version, suggesting that a close read and review has not yet been completed across the full manuscript. However, certain sections of the manuscript are particularly well-written.

Section “Deglacial [CO₃₂-] reconstruction” should be revised to offer a basic process understanding to the reader of what higher or lower reconstructed [CO₃₂-] values mean. Reiterate for each core what the combined influence of the respective water masses have on [CO₃₂-]. Why do the deeper cores have higher CO₃₂- during the LGM when C should be highly accumulated? The reader has to work too hard in this section to make sense of your results as written.

Section “Factors influencing” I would suggest reordering your paragraphs to discuss first your proxy records that support ventilation being the chief factor. This is more in line with the papers discussion thus far and also emphasizes importance. Then turn to consider that productivity could also potentially have an influence, but that your proxy results examining bio pump strength suggest otherwise.

Abstract – Editor input/confirmation needed, abbreviations in the abstract should be removed and revisions are desired to improve the readability by a broad audience

Line 27 “ca. 40” the use of ca. should be removed from the entire body of the manuscript and the more traditional use of ~ (which is also used in some cases herein) or the word approximately should be used.

Line 42 replace “has yet to emerge” with “is needed” for better flow

Line 43-46 suggest revise sentence, for example “The Southern Ocean likely played an important role in deglaciation, given its ability to connect the deep oceanic reservoir and the atmosphere via surface exchange processes.” Is this a place where you can also nod specifically to the Pacific Southern Ocean and its importance? Also, is there a seminal reference on Southern Ocean influence in the global C cycle that you can include here for an overview for readers?

Line 47 replace “by” with “using”

Line 47 begin sentence with LGM to orient the reader to your reconstruction time period

Line 47 you introduce the utility of $\delta^{13}C$ below (line 74), but not here where it’s first mentioned. There is also no explanation to what Cd/Ca is a proxy for – if you mention these proxy by name, you should provide some context, alternatively you could revise to say something more vague like “paleoclimate proxy reconstructions have shown... with your references cited there for the reader to investigate further if they’d like.

Line 47 is it relevant/important to include where these records were generated i.e. “benthic foraminifera collected in the XXX”

Line 48 period after MOC, start new sentence with “Overall, this slowed physical ocean movement resulted in increased nutrient and carbon concentrations in the deep ocean, particularly within glacial deep Pacific and Southern Ocean, resulting in isolation of these biogeochemical constituents from the surface ocean and atmosphere during the LGM.” (inclusion of surface ocean since nutrients are mentioned).

Lines 50-52 with the revision above, this is now largely redundant. Suggest moving references to prior sentence and eliminating this sentence

Line 53 I would suggest eliminating “to the edge of the modern winter sea ice maximum” for brevity/clarity in this long sentence

Line 56 put “below ~2000 m in parenthesis and eliminate water depth

Line 57 suggesting replacing “generally” with “relatively” if the point is to compare glacial-interglacial conditions. Is “well” needed in this statement about ventilation?

Line 58 revise “,thereby conditioning...” to “, resulting in additional deep-water carbon accumulation in already old, carbon-rich Pacific and Southern Ocean deep waters. Previous studies and our results suggest these source waters played a pivotal role in ocean carbon ventilation (or efflux) in the subsequent deglaciation.

Lines 60-72 this paragraph is complex, full of abbreviations, and difficult to follow. Revise for clarify and distill this information to the most critical water masses (AABW and PDW) that are important to your

“story”... I know word limitations are hindering, but avoiding semi-uncommon abbreviations (like these water mass names) as much as possible is ideal.

Line 73 revise to “Most paleoclimate interpretations of MOC and...”

Line 76 this point is really important to emphasizing the novelty of your study, glad you made it

Line 78 replace “essential” with “is a means to estimate deep ocean carbon storage”

Line 81 end sentence following mostly where Yu et al is cited. Eliminate or soften the language in the rest of this sentence. There has been a body of work focused on this proxy and my read of this statement is misleading and undermines this tool. Reference 43 is an example of a successful application of this tool that should also be cited following reference 23, there are many others but I know # of citations is also limiting.

Line 80 This would be a good place to introduce $\delta[\text{CO}_3^{2-}]$, which is not defined

Line 92 reference needed.

Line 96 area density is a conventional proxy for surface ocean $[\text{CO}_3^{2-}]$, which can be influenced by dissolution intensity in under-saturated water. However, the secondary process is much more challenging to quantify. This and the subsequent sentence are unclear and perhaps misleading as written, and need revision.

Line 97 Revise to “This nature of the area density proxy poses challenges to using it as a tool to quantitatively reconstruct deep-water ...” remove “is difficult” at end of sentence as written.

Line 104 HOW?

Line 109 (Figure 1)

Line 115 eliminate “in the Southeast Pacific” redundant

117 can changes be revised to provide more process information about what is happening with pCO_2 ? The inclusion of “, along atm pCO_2 changes” is unclear.

Line 121 eliminate (%Low-CT-number calcite volume) until you get into your body enough to contextualize what this actually means

Line 123 following “wall thickness” (which is indicative of surface ocean $[\text{CO}_3^{2-}]$ during foram calcification) to clarify process to the reader.

Line 121-124 this is a lot of information and a long sentence, I suggest breaking it into 2

Line 127 (Figure 2a). Start new sentence. This indicates that...

Line 135-136 can you clearly state here or elsewhere, your resulting $[\text{CO}_3^{2-}]$ estimates from your new proxy approach, are these estimates for CO_3^{2-} at the sediment-water interface/seafloor? If yes, it might be useful to use a subscript with your mentions of $[\text{CO}_3^{2-}]_b$ to indicate bottom water or something like that.

Line 142 what assumptions are made about α in your reconstructions where these values are unknown? Is this also taken into consideration in your error reported on line 145?

Line 145 define how you estimated your reported uncertainty, is this a fully propagated error or only the error associated with the regression?

Line 158 really interesting to read that your cores resolved the foram barren interval, how long did this persist in your record approximately?

Line 169 typo "a mixture", eliminate "from below", redundant

Line 175 Capitalize start of new sentence and eliminate "which likewise"

Line 176 Delete "be caused by"

Line 178 mention of Zr/RB requires you to provide context, orient your reader to what this tool actually is a measure for.

Line 181-182 Delete "while it was the..." and onward.

Line 184 eliminate "thus"

189-190... this is absolutely a factor influencing deep water [CO₃²⁻] across modern and paleo time-scales, revise to emphasize that you are investigating the significance of this process.

Line 193 Br/Ca of bulk sediments?

Line 220 typo... which "can be"

Line 231 I'm not sure what a "biogeochemical configuration" is. Suggest eliminating/revising, and merging with following sentence... "Deep-water [CO₃²⁻] structure can be altered by changes in deep-water circulation and/or changes in DIC content of water masses, which occur largely as a result of exchange with the atmospheric C reservoir."

Line 234 Eliminate "However" and remove comma on line 235

Line 242-243, this was already stated above, eliminate redundancies

Line 271 Remove "In the following" this is an example of extra words that can be cut from your text for clarity/brevity. (Also Line 272 eliminate "Firstly")

Lines 271-308 This section was very well-written, really interesting results and great use of other studies for comparison to understand your data

Lines 312-331 This is important introductory information, which would better serve your reader by being presented around Line 60 of your submitted manuscript. Subsequent discussion of your data in relation to this information should remain here.

Line 386 Eliminate this sentence, it' is redundant with line 381 and begin a new paragraph

Line 387 Remove "in fact, however" and replace with "Research has shown"

Line 389-392 shorten and merge these two results and datasets into one sentence “Previous studies suggest that during glacial periods, deep Southern Ocean alkalinity was $\sim 25 \mu\text{mol kg}^{-1}$ higher (refs) and estimate alkalinity values of $\sim 300 \mu\text{mol kg}^{-1}$ (refs).

Line 394 Redundant, delete “than under the assumption of stable alkalinity”

Line 399 replace “indispensable” with critical or important

Reviewer #2 (Remarks to the Author):

The paper by Iwasaki et al. represents an early attempt to use the nascent X-ray Micro CT method to interpret bottom water carbonate ion data from the density of foraminiferal shells. Here, records from the SE Pacific margin are used to interpret the evolution of intermediate-deep water masses over the last deglacial interval. Results highlight the roll of the biological pump and deep water carbon sequestration, especially in the Pacific sector of the Southern ocean in glacial carbon storage.

The paper is well written and presents a very reasonable case study on which to apply this emerging proxy. However, I do have some concerns about the degree to which this rather new method is being used to interpret broad-scale circulatory changes. I actually find all of the interpretations fundamentally plausible, but largely because of their support within the existing framework for deglacial carbon storage. And while this paper would be a valuable contribution on these grounds alone, the truly novel piece here is in the robust application of X-ray microCT. This is where I, and likely others readers, may need some additional convincing that this method is ready for downcore use. I hope that my critical comments on this point are helpful.

I have three points of concern around the use of X-ray microCT as a proxy as applied here.

1. Sample size

The use of 8 shells at a 20cm resolution is a surprisingly small sample size from which to generate a record of $[\text{Co}^{32}]$. I understand that these analyses may be quite labor intensive, however, that alone cannot justify the use of such a small population. Especially without giving an indication of the amount of within-population variability or error of individual measurements. Results would be far more robust if the use of such a small sample size could be quantitatively justified and/or increased.

I will note that it is not just the sample sizes of X-ray microCT samples that are quite small, but the use of only 5 *G. bulloides* for isotope analyses is also surprisingly small. Please see Fraass and Lowery (2017) for a thorough discussion of sample size, but where possible, closer to 20 shells is generally considered prudent.

2. Selection of “unblemished” shells

a. The authors describe handpicking “unblemished” shells for X-ray microCT analyses. The phrase may require some clarification, but I interpret this to mean shells that appeared pristine under a light microscope. However, would this not bias selection towards shells that had different (lesser) degrees of dissolution? How degree of dissolution as measure by X-ray microCT may or may not correspond to appearance under a light microscope is an important missing piece of information!

3. The claim of disassociation between shell thickness (surface processes) and density (bottom water).

a. The claim is made that while surface processes may impact SNW through shell thickness, X-ray microCT density measurements should be largely reflective of bottom water saturation. I think the distinction between density as a proxy for bottom water saturation and shell thickness for surface processes is broadly supported by both the authors’ previous work and the latitudinal transect of core tops presented here. However, it strikes me that these two factors may not be truly independent. For example, a thinner shell may be more prone to dissolution if only due to an increase in surface area:volume of calcite. We know that CO₂ and thus near-surface [CO₃²⁻] likely changed over the deglaciation. Moreover, this region may (?) be within the influence of active upwelling along the Chile margin, which may have been variable in intensity and carbonate chemistry through the last deglaciation. Thus, how can one rule out or disentangle potential impacts of calcification (surface saturation) on subsequent dissolution (bottom water saturation)? This is not meant to be a challenge, but a genuine question I am left with.

4. Novel proxy would be more robust with a supporting record

a. I think the greatest weakness here is really the remaining uncertainties around this proxy, some of which are articulated above. And, while I realize this would imply substantial additional work, I’d urge the authors to consider adding one or more supporting records utilizing a more conventional proxy. The obvious choice would be a cibicoides (or other benthic foraminiferal) B/Ca (or d₁₁B) record. Doing so would allow for a more robust direct comparison with the Atlantic records in question. And if carried out in contemporaneous samples, could provide a much-needed test of how this emerging density proxy functions downcore.

In addition to this, I find interpretations of X-ray microCT ([Co₃²⁻]) data lacking in two areas.

1. Error bars associated with X-ray microCT based [Co₃²⁻] are shown in Figure 3 but not defined. What do these refer to? Do these include propagated error from both measurements and the calibration? It’s a little hard to interpret without understanding what these bars represent, but I will note that most points seem to be “within error” of one another. If this is the case, it may be worthwhile to be more explicit (and cautious) in limiting interpretations to signals that appear truly robust. I also wonder if this issue could not be improved with an increase in sample size.

2. Calculations of DIC are based on unrealistic assumptions.

Using modern conditions here does not seem appropriate. In addition to G-IG changes in TA as discussed, TA could potentially be influenced by both changes in surface production and export (e.g., associated with changes in carbonate export as discussed by the authors), as well as changes in

dissolution. Temp and salinity are potentially more defensible but would require additional evidence that the water mass being recorded at each depth has been consistent. The same caveat would also apply to TA. If the authors are arguing for a change in influence of Northern (PDW) vs Southern sourced waters in some intervals, however, this becomes difficult.

I'd strongly suggest authors a) recalculate, propagating errors from a full range of relevant potential TA, T, and S values; b) seek additional constraints on parameters - perhaps using model results, external estimates of which I'm not aware, or re-evaluating their own data. For example, if authors collected B/Ca data, they may include Mg/Ca in their analytes which could further constrain temperature. OR c) acknowledge that sufficient constraints may not exist to calculate DIC and rely on [CO₃²⁻].

I've also added a few minor line-by-line points below.

- Could you label UCDW and LCDW on your Fig 1?

- Line 83: There have been huge advances here in understanding the mechanistic relationships between B/Ca and carbonate ion in the past decade or so - I'd recommend the book Boron Proxies in Paleoceanography and Paleoclimatology (and of course references therein) as an overview.

- Line 135: Do you mean CaCO₃ saturation? Otherwise, why include the assumption of Ca consistency?

- 168: What is the test of significance used?

- 192: I'm assuming here and elsewhere this is a typo and you mean Ba/Ca (or possibly B/Ca?). If this is not a typo, additional explanation of the utility of Br in this context is certainly warranted. In either case there is no discussion of sedimentary (I assume this is sedimentary and not from foraminifera?) Ba or Br analyses. This should be added to the methods.

- Figure 3: Consider making the y axes comparable across these 4 plots as it is very difficult to compare magnitude of change at the different depths as currently plotted.

- Figure 3: Also, why do line plots continue past the last plotted point in b and c?

- Figure 3e&f: please include units.

Reviewer #3 (Remarks to the Author):

The manuscript by Iwasaki et al. presents new data of X-ray Micro-Computer-Tomography of planktonic foraminifera *G. bulloides* to reconstruct deglacial changes of deep water CO₃²⁻ along a depth transect of the South-East Pacific. The method is recent and is applied for the first time for this area.

The wide Pacific Ocean is not as well documented as the Atlantic Ocean and these new data are of interest. However, both results description and discussion are confusing and should be better organized.

While the paper aim at describing the oceanic circulation changes between the Pacific and Atlantic sector of the Southern Ocean, it does not take into account the fact that throughout the deglaciation, the cores studied might not be bathed by the same water masses, for example it might be the case for PC01 bathed by AAIW during different periods of the deglaciation as it has been published both for the Southwest and Southeast Pacific (Ronge et al., 2015; Haddam et al., 2020). The assumptions made on the Last Glacial Atlantic circulation could also be opposite to previous publications and this is not discussed (see comment Line 261-263). The authors compare their data with some published data from the Southwest, central and Southeast Pacific but some of the most recent publications are missing. The author should also compare their results with pH reconstructions for the deep waters of the Drake passage from $\delta^{11}\text{B}$ measured in deep sea corals (Rae et al., 2018 and Li et al., 2020) as they consider the influence of Pacific deep waters supplied to the Atlantic Ocean via the Drake Passage.

I think the discussion should be completely revised before considering this manuscript for publication.

Other comments/questions:

Line 37: if you consider preindustrial interglacial, rounding up to 60 would be more appropriate

Line 48: $\delta^{13}\text{C}$ and Cd/Ca indicates water masses geometry, not the strength of the flow. Like for air bubbles ice core CO₂ papers you could cite older publications (Curry et al., 88 Duplessy et al., 88,..)

Line 50 “in particular” why ? Deeper gradient have been observed for Atlantic and Indian Ocean also (for $\delta^{18}\text{O}$ for example).

Line 55: could cite also Stephens and Keeling 2000, Ferrari et al., 2014...

Line 63: Following Talley 2013 the salinity minimum characterizing AAIW extend deeper than 1000 m, 1300-1400 m would clearly be more appropriate.

Line 64: Mainly NADW? Following Talley 2013, the contribution to CDW are 9 Sv from the Pacific Ocean, 16 Sv from the Indian (from which 5Sv comes directly from NADW), and 13 Sv from the Atlantic, thus at maximum 18Sv from the Atlantic and 20 from the Indian and Pacific oceans.

Line 69: relatively more influenced by NADW and AABW characteristics

Line 82: relationship between bottom water ΔCO_2 - and dissolution ID_X (μCT) is also empirical

Line 90: In Allen et al. 2019 that is cited, B/Ca has been measured for two cores (1627 m, 2541 m) that are influenced partly influenced by UCDW.

Line 114: sections presenting CO₂- concentrations would be nice also. AAIW would be better represent on a salinity section. As the cores are closer to the continent than the section represented, PCO₂ core is probably bathed by PDW and not CDW nowadays.

Line 137: temperature and salinity of deep water are clearly not stable throughout the deglaciation, would be nice that the authors estimate the error corresponding to changes in salinity and possible changes in temperature.

Line 145: Are both uncertainties linked to the empirical relationship and analytical determination %Low CT number calcite volume both taken into account?

Line 152: the shallowest site is bathed nowadays by PDW waters, and by AAIW during the warming events of the deglaciation (Haddam et al., 2020)

Line 156: PCO₂ bathed by PDW nowadays, see previous comment.

Line 161-162: hard to follow: PCO₂ CO₃²⁻ values from 19 to 17 ka indicate a sharp increase from 70 to ~90 μmol/kg, followed by a sharp decrease to values ~80 μmol/kg, with a rather stable period until 15 ka, followed by a decrease to 67 μmol/kg between 15 and 14 ka.

Line 164: The first decrease in CO₃²⁻ for core PCO₂ between 21 and 20 ka is not observed for PCO₃ but maybe the authors sentence "similar variations" is linked to values 82-75 μmol/kg of PCO₃ for the interval barren of foraminifera in core PCO₂?

Line 166-167: again difficult to follow: from 19 to 15 ka both cores indicates higher CO₃²⁻ values at the beginning (~90 μmol/kg) and a decrease to lower values after. For the end of that period the different resolution makes it difficult to compare the two core records.

Line 173-178: long sentence, not very clear. What is the impact of the strong bottom current flow?

Line 196, fig 3e: PCO₂ indicate higher Br/Ca value at the end of the LGM than during 19-15 ka. For PCO₃ the values are more similar. It does not correspond to the authors description. Furthermore the comparison with Br/Ca of publication 40 might be unfounded: the authors from this publication argued that the high sedimentation rate of their core and the shallow depth favoured the preservation of organic matter (and Br) and carbonates and thus could be linked to the fluxes escaping the surface.

Line 220 "can be"?

Line 242-246: The authors present PCO₁ as bathed by AAIW and UCDW on lines 152 and by PDW here. They do not discuss their reconstructed CO₃²⁻ record compared to Haddam et al, 2020 (and Ronge et al., 2015) indicating that during early deglaciation AAIW water occupied deeper depth and notable bathed PCO₁. This changing water mass could correspond to the CO₃²⁻ increase observed from ~16.5 to 15 ka.

Lines 250-251: CO₃²⁻ record from core PS75/054-1 indicate variations from 85 to 100 μmol/kg that is not "largely invariant and rather of similar amplitude than variations from ~80 to 95 μmol/kg from core PCO₃.

Line 261-263: As explained before by the authors, GNAIW was shallower and a larger flux at shallow depth does not imply a larger influence on CDW during the last glacial maximum. As proposed by Keeling and Stephens, 2001, Ferrari et al., 2014 among others, the deep ocean circulation probably correspond to a deep overturning cell disconnected from the shallow overturning GNAIW cell.

Line 272-274: seems to be opposite to line 250-253

Line 300 suppress “that”

Line 302-304: the authors could also comment on the fact that different tracers were used?

Line 317-330: main remark, the authors should consider deep sea corals $\delta^{11}\text{B}$ reconstructions from the Drake passage (Rae et al., 2018, Li et al., 2020).

Line 337-338: Productivity is thought to have been lower at the LGM in Antarctic surface waters, on the opposite, LGM productivity of subantarctic waters was enhanced, probably due to iron availability (Martin et al., 1990, Martinez-Garcia et al., 2011, 2014 among a number of other papers..)

Line 389-390: clearly not the good references for deep waters alkalinity changes.

Line 393 -394 suppress “also”

Material and methods:

Supplemental fig 3a and 3b are referenced first in the text, corresponding to figure suppl 2a and 2b and fig. suppl. 1 is referenced after. Supplemental fig. 2c should be referenced line 450-453. The DIC calculations should be presented in the material and method part.

Figures: it would help to have figures numbered

Figure 1: add c line 566 for 30°S section.

Figure 4a: the authors should use the atmospheric $\delta^{13}\text{C}$ record of Bauska et al., 2016.

Figure 5c: the vertical axis is wrong.

Figure 6b: the carbonate counter pump will reduce the efficiency of the soft tissue pump, not “discharge CO_2 to the atmosphere”. Resumption of the Southern Ocean upwelling will conduct to ocean CO_2 degassing.

In the following, we have compiled our specific replies to the reviewers' comments. The replies are arranged in a following manner:

1. *Reviewer's comments;*
2. **Our reply and actual sentence/phrase/paragraph we wrote and placed in text .**

Response to Reviewer #1:

(comments by the reviewer are printed in italics, our response in red)

Replies back to Reviewer #1 comments:

General comments:

The submitted manuscript builds nicely on a body of paleoclimate research that attempts to answer the important question of the source and process for CO₂ outgassing associated with the glacial-interglacial transition. The authors develop a new proxy tool that is uniquely suited to investigate deep water carbonate chemistry dynamics and apply this tool to a transect of cores in the Pacific Southern Ocean, a region that is likely a critical zone for ultimate ocean-atmospheric CO₂ release processes. The authors present compelling results and extensively relate their results to other studies within the region, which strengthen the interpretation of their results. Most excitingly, the authors (using some basic assumptions) are able to quantitatively estimate the fraction of CO₂ (in GtC) released from the Pacific Southern Ocean. This exciting work brings together decades of research on this particular topic and region and blends multi-proxy approaches to understand the dynamics and physical processes at play. The results of this work are well suited for Nature Communications.

Specific section and line by line comments are below.

Overall: Revisions are needed for clarity, distill the important information, include important process information when discussing proxy results and eliminate abbreviations where possible. Stylistically, review to eliminate any unnecessary words to overall work towards shortening your sentences, breaking long sentences in two when needed. This will make the manuscript much easier to read and follow. Basic typos still exist within this final version, suggesting that a close read and review has not yet been completed across the full manuscript. However, certain sections of the manuscript are particularly well-written.

Specific comments and suggestion:

Section "Deglacial [CO₃₂-] reconstruction" should be revised to offer a basic process understanding to the reader of what higher or lower reconstructed [CO₃₂-] values mean. Reiterate for each core what the combined influence of the respective water masses have on [CO₃₂-]. Why do the deeper cores have higher CO₃₂- during the LGM when C should be highly accumulated? The reader has to work too hard in this section to make sense of your results as written.

Following the suggestions, we revised the description to make it easier to understand. We added notes on the respective deep-water masses on each sample site to enable a better understanding which water-mass each sample site represents, and added an annotation of DIC states in the figure to show what $[\text{CO}_3^{2-}]$ change means. Please see the first paragraph of this section in the revised version.

Section “Factors influencing” I would suggest reordering your paragraphs to discuss first your proxy records that support ventilation being the chief factor. This is more in line with the papers discussion thus far and also emphasizes importance. Then turn to consider that productivity could also potentially have an influence, but that your proxy results examining bio pump strength suggest otherwise.

Thank you for the helpful suggestions. in the revised version, we have rearranged the order to the paragraph as advised by the reviewer. Please see the revised section “Factors influencing $[\text{CO}_3^{2-}]$ changes..”

Abstract – Editor input/confirmation needed, abbreviations in the abstract should be removed and revisions are desired to improve the readability by a broad audience

The abbreviations were removed from the revised Abstract.

Line 27 “ca. 40” the use of ca. should be removed from the entire body of the manuscript and the more traditional use of ~ (which is also used in some cases herein) or the work approximately should be used.

Following the suggestions, we revised the entire manuscript.

Line 42 replace “has yet to emerge” with “is needed” for better flow

Thank you. We corrected it.

Line 43-46 suggest revise sentence, for example “The Southern Ocean likely played an important role in deglaciation, given its ability to connect the deep oceanic reservoir and the atmosphere via surface exchange processes.” Is this a place where you can also nod specifically to the Pacific Southern Ocean and its importance? Also, is there a seminal reference on Southern Ocean influence in the global C cycle that you can include here for an overview for readers?

Following the suggestion, we revised the sentences as follows:

Line48: The Southern Ocean, in particular the large Pacific sector, likely played an important role in deglaciation, given its size and ability to directly connect the deep oceanic reservoirs with the

atmosphere via surface exchange processes⁵⁻¹⁰.

Line 47 replace “by” with “using”

Following the suggestions, we revised the manuscript.

Line 47 begin sentence with LGM to orient the reader to your reconstruction time period

Following the suggestions, we revised the manuscript.

Line 47 you introduce the utility of $d13C$ below (line 74), but not here where it's first mentioned. There is also no explanation to what Cd/Ca is a proxy for – if you mention these proxy by name, you should provide some context, alternatively you could revise to say something more vague like “paleoclimate proxy reconstructions have shown... with your references cited there for the reader to investigate further if they'd like.

We followed the suggestion. The sentences were revised as follows:

Line42: During the LGM, most paleoclimate proxy reconstructions indicate a generally shallower Atlantic MOC^{11, 12}. Overall, this slowed physical ocean movement resulted in increased nutrient and carbon concentrations in the deep ocean, particularly within glacial deep Pacific and Southern Ocean, resulting in isolation of these biogeochemical constituents from the surface ocean and atmosphere during the LGM¹¹⁻¹⁵.

Line 47 is it relevant/important to include where these records where generated i.e. “benthic foraminifera collected in the XXX”

The description was revised as suggested in the above comment.

Line 48 period after MOC, start new sentence with “Overall, this slowed physical ocean movement resulted in increased nutrient and carbon concentrations in the deep ocean, particularly within glacial deep Pacific and Southern Ocean, resulting in isolation of these biogeochemical constituents from the surface ocean and atmosphere during the LGM.” (Inclusion of surface ocean since nutrients are mentioned).

Following the suggestions, we revised the manuscript as follows:

Line 53: Overall, this slowed physical ocean movement resulted in increased nutrient and carbon concentrations in the deep ocean, particularly within glacial deep Pacific and Southern Ocean, resulting in isolation of these biogeochemical constituents from the surface ocean and atmosphere during the LGM¹¹⁻¹⁵.

Lines 50-52 with the revision above, this is now largely redundant. Suggest moving references to prior sentence and eliminating this sentence

We followed the suggestion. The sentence was eliminated.

Line 53 I would suggest eliminating “to the edge of the modern winter sea ice maximum” for brevity/clarity in this long sentence

Line 56 put “below ~2000 m in parenthesis and eliminate water depth

Line 57 suggesting replacing “generally” with “relatively” if the point is to compare glacial-interglacial conditions. Is “well” needed in this statement about ventilation?

We followed these suggestions and corrected manuscript.

Line 58 revise “,thereby conditioning...” to “, resulting in additional deep-water carbon accumulation in already old, carbon-rich Pacific and Southern Ocean deep waters. Previous studies and our results suggest these source waters played a pivotal role in ocean carbon ventilation (or efflux) in the subsequent deglaciation.

We followed the suggestion. Manuscript was revised as follows:

Line 61: resulting in additional carbon accumulation in already carbon-rich, old Pacific and Southern Ocean deep waters. Previous studies hypothesized that such old deep-water in the Pacific and Southern Ocean became a source of carbon efflux in the following deglaciation.

Lines 60-72 this paragraph is complex, full of abbreviations, and difficult to follow. Revise for clarity and distill this information to the most critical water masses (AABW and PDW) that are important to your “story”... I know word limitations are hindering, but avoiding semi-uncommon abbreviations (like these water mass names) as much as possible is ideal.

We revised this paragraph so that make it easier to follow.

Line 70: Today, in the subantarctic South Pacific around 45°S, the water mass structure deeper than 1000 m is broadly characterized by contributions of four principal water masses; Antarctic Intermediate Water (AAIW: ~1000 m), which flows northward, Pacific Deep Water (PDW: ~1000-2000 m) flow southwards, Antarctic Bottom Water (AABW: below ~4500 m), which moves northwards, and aged, modified North Atlantic Deep Water (NADW), which is transported by the Antarctic Circumpolar Current (ACC) from the Atlantic²⁹. The latter is hence commonly re-defined as Circumpolar Deep Water (CDW), and occupies a broad water depth range between 2000-4000 m depth, leading to varying degrees of mixing with the neighboring upper and lower water masses. As a result, modern CDW is divided into Upper CDW (UCDW), relatively more influenced by AAIW and PDW characteristics, and Lower CDW (LCDW), relatively more influenced by AABW and aged NADW characteristics (Figure 1).

Line 73 revise to “Most paleoclimate interpretations of MOC and...”

Line 78 replace “essential” with “is a means to estimate deep ocean carbon storage”

We followed the suggestion and corrected.

Line 81 end sentence following mostly where Yu et al is cited. Eliminate or soften the language in the rest of this sentence. There has been a body of work focused on this proxy and my read of this statement is misleading and undermines this tool. Reference 43 is an example of a successful application of this tool that should also be cited following reference 23, there are many others but I know # of citations is also limiting.

Following the suggestions, we revised the description and added new references as follows:

Line 89: The B/Ca ratio of epifaunal benthic foraminifera, particularly *Cibicides wuellerstorfi*, has so far been the most-often deep-water [CO₃²⁻] proxy, developed on the basis of empirical correlation with the deep-water carbonate saturation state (Δ [CO₃²⁻])^{25, 30, 31}.

Line 80 This would be a good place to introduce delta [CO₃²⁻], which is not defined

Following the suggestion, we defined Δ [CO₃²⁻] in this paragraph.

Line 92 reference needed.

We added a new reference to this sentence.

Line 96 area density is a conventional proxy for surface ocean [CO₃²⁻], which can be influenced by dissolution intensity in under-saturated water. However, the secondary process is much more challenging to quantify. This and the subsequent sentence are unclear and perhaps misleading as written, and need revision.

As the reviewer suggested, the area density is generally recognized as a proxy of sea-surface condition. In the revised manuscript, we employed the values of “SNW”, which has been used as both indicator of bottom and surface water. The sentences were revised to answer the suggestion.

Line 106: The size-normalized weight (SNW) of planktic foraminiferal tests is a conventional proxy of carbonate dissolution intensity in under-saturated water^{34,35}. However, the surface water carbonate chemistry significantly affects the calcification intensity of planktic foraminifera and alters their test thickness prior to carbonate dissolution after death and deposition at the seafloor, thus can affect the SNW proxy as well³⁶⁻³⁸. This nature of the SNW proxy poses challenges to using it as a tool to quantitatively reconstruct deep-water [CO₃²⁻] in the areas, where glacial-interglacial sea surface environmental variation was large.

Line 97 Revise to “This nature of the area density proxy poses challenges to using it as a tool to quantitatively reconstruct deep-water ...” remove “is difficult” at end of sentence as written

We followed the suggestion and corrected.

Line 104 HOW?

We added the sentences to describe how we overcame the weak point as follows:

Line 117: Focusing on the micro-scale density distribution in specimen, this new proxy is not affected by geometric characteristics of test like size or thickness, which are influenced by sea surface conditions.

Line 109 (Figure 1)

Line 115 eliminate “in the Southeast Pacific” redundant

We followed the suggestion and corrected.

Line 121 eliminate (%Low-CT-number calcite volume) until you get into your body enough to contextualize what this actually means

Following the suggestion, we rearranged the manuscript.

Line 123 following “wall thickness” (which is indicative of surface ocean [CO₃2-] during foram calcification) to clarify process to the reader.

The detail of relationship between test wall thickness and sea surface condition was described in the Introduction. Accordingly, the sentence was revised as following.

Line 140: It enables us to exclude the effects of wall thickness, which are changing with sea surface conditions, and therefore provides a reliable method to evaluate the dissolution intensity of this species' tests.

Line 121-124 this is a lot of information and a long sentence, I suggest breaking it into 2]Line 127 (Figure 2a). Start new sentence. This indicates that...

Following the suggestion, we revised the manuscript.

Line 135-136 can you clearly state here or elsewhere, your resulting [CO₃²⁻] estimates from your new proxy approach, are these estimates for CO₃²⁻ at the sediment-water interface/seafloor? If yes, it might be useful to use a subscript with your mentions of [CO₃²⁻]_b to indicate bottom water or something like that.

Following the suggestion, we revised the manuscript as following:

Line 147: Calibration between the dissolution index, defined as %Low-CT-number calcite volume in this study, and deep-water $\Delta[\text{CO}_3^{2-}]$ at each core site in the Southeast Pacific shows that the %Low-CT-number calcite volume is effective as a quantitative proxy of deep-water $\Delta[\text{CO}_3^{2-}]$, precisely at the bottom water – sediment interface for the study region (Figure 2b).

Line 142 what assumptions are made about a in your reconstructions where these values are unknown? Is this also taken into consideration in your error reported on line 145?

In the revised manuscript, we also described the impact of glacial-interglacial change into this constant value. And as the impact is not significant, we used the modern value. The revised sentence is as following:

Line 160: The $[\text{CO}_3^{2-}]_{\text{sat}}$ is a constant depending on the bottom water temperature, salinity, and pressure at each core site. The overall effects of these parameters in the deep ocean $[\text{CO}_3^{2-}]_{\text{sat}}$ on glacial-interglacial time scales change about $0.5 \mu\text{mol kg}^{-1}$, based on $\sim 3^\circ\text{C}$ change in bottom water temperature, ~ 1.5 psu change in salinity, and ~ 120 m equivalent change in pressure⁴². Therefore, modern $[\text{CO}_3^{2-}]_{\text{sat}}$ values are employed to calculate down-core $[\text{CO}_3^{2-}]$.

Line 145 define how you estimated your reported uncertainty, is this a fully propagated error or only the error associated with the regression?

We also described about error in the revised manuscript as following:

Line 165: Based on the standard error of individual % Low-CT-number calcite volume measurement in each sample and our established calibration using multiple core samples, the uncertainty associated with reconstructing deep-water $[\text{CO}_3^{2-}]$ is $\sim 5.0 \mu\text{mol kg}^{-1}$ at the 95% confidence level. This makes it possible to detect $\sim 10 \mu\text{mol kg}^{-1}$ variations of $[\text{CO}_3^{2-}]$ on millennial or longer time-scales.

Line 158 really interesting to read that your cores resolved the foram barren interval, how long did this persist in your record approximately?

Foram barren interval is about 1000 years at the end of LGM (19 ka). The sentence was revised as following:

Line 181: During the LGM (19-21 ka BP), PCO₂ (UCDW) shows values between 77-90 $\mu\text{mol kg}^{-1}$, followed by a short, ~ 1000 year foraminifera-barren interval at the end of LGM (19 ka BP),

supposedly driven by a carbonate dissolution event due to low $[\text{CO}_3^{2-}]$ deep water intrusion.

Line 169 typo “a mixture”, eliminate “from below”, redundant

Line 175 Capitalize start of new sentence and eliminate “which likewise”

Line 176 Delete “be caused by”

Following the suggestions, we revised the manuscript.

Line 178 mention of Zr/Rb requires you to provide context, orient your reader to what this tool actually is a measure for.

Data and discussion of Zr/Rb was eliminated from revised manuscript.

Line 181-182 Delete “while it was the...” and onward.

Line 184 eliminate “thus”

Following the suggestions, we revised the manuscript.

Line 189-190... this is absolutely a factor influencing deep water $[\text{CO}_3^{2-}]$ across modern and paleo time-scales, revise to emphasize that you are investigating the significance of this process.

Here we intend to highlight the local effect of organic material decomposition that alter $[\text{CO}_3^{2-}]$ in sediment surface. In order to make it clear, we rearranged the manuscript as follows.

Line 225: On the other hand, organic matter burial and remineralization in surface sediments is an alternative factor that locally alters the values of porewater $[\text{CO}_3^{2-}]$ in the sediment and carbonate dissolution intensity in the sediment surface. On the Chilean margin, variations in the bulk sediment Br/Ca ratios serve as an indication of the biological carbon pump efficiency, based on the fact that in this area sedimentary bromine (Br) and calcium (Ca) are closely associated with biogenic organic carbon and carbonate content, respectively⁴⁷. In our study, there appears no apparent link between the variation in Br/Ca ratios and carbonate preservation at the bathyal sites PC02 and PC03 (Figure 3f). Although, local differences exist in surface productivity or depth-dependent differences in the carbonate and organic carbon preservation at our bathyal sites, the discrepancy between the Br/Ca ratio and carbonate dissolution patterns suggests that the organic carbon burial and decomposition in the sediment surface is not a principal controlling factor of carbonate dissolution intensity at our study sites. Therefore, we consider the carbonate chemical condition in deep-water mass to be principally reflecting a true water mass signal unaltered by porewater chemistry changes.

Line 193 Br/Ca of bulk sediments?

Line 220 typo... which “can be”

We followed the suggestion and corrected.

Line 231 I’m not sure what a “biogeochemical configuration” is. Suggest eliminating/revising, and merging with following sentence... “Deep-water [CO₂] structure can be altered by changes in deep-water circulation and/or changes in DIC content of water masses, which occur largely as a result of exchange with the atmospheric C reservoir.”

Thank you for the helpful suggestion, we revised the manuscript based on this comment.

Line 234 Eliminate “However” and remove comma on line 235

Line 242-243, this was already stated above, eliminate redundancies

Line 271 Remove “In the following” this is an example of extra words that can be cut from your text for clarity/brevity. (Also Line 272 eliminate “Firstly”)

We followed the suggestions and corrected.

Lines 312-331 This is important introductory information, which would better serve your reader by being presented around Line 60 of your submitted manuscript. Subsequent discussion of your data in relation to this information should remain here.

We followed the suggestion. We moved this into the Introduction.

Line 386 Eliminate this sentence, it’ is redundant with line 381 and begin a new paragraph

Line 387 Remove “in fact, however” and replace with “Research has shown

Line 389-392 shorten and merge these two results and datasets into one sentence “Previous studies suggest that during glacial periods, deep Southern Ocean alkalinity was ~25 $\mu\text{mol kg}^{-1}$ higher (refs) and estimate alkalinity values of ~300 $\mu\text{mol kg}^{-1}$ (refs).

Line 394 Redundant, delete “than under the assumption of stable alkalinity”

Line 399 replace “indispensable” with critical or important

We followed all suggestions, and corrected the manuscript.

Response to Reviewer #2:

(comments by the reviewer are printed in italics, our response in red)

back to Reviewer #2 comments:

General comments:

The paper by Iwasaki et al. represents an early attempt to use the nascent X-ray Micro CT method to interpret bottom water carbonate ion data from the density of foraminiferal shells. Here, records from the SE Pacific margin are used to interpret the evolution of intermediate-deep water masses over the last deglacial interval. Results highlight the roll of the biological pump and deep water carbon sequestration, especially in the Pacific sector of the Southern ocean in glacial carbon storage.

The paper is well written and presents a very reasonable case study on which to apply this emerging proxy. However, I do have some concerns about the degree to which this rather new method is being used to interpret broad-scale circulatory changes. I actually find all of the interpretations fundamentally plausible, but largely because of their support within the existing framework for deglacial carbon storage. And while this paper would be a valuable contribution on these grounds alone, the truly novel piece here is in the robust application of X-ray microCT. This is where I, and likely others readers, may need some additional convincing that this method is ready for downcore use. I hope that my critical comments on this point are helpful.

Specific comments and suggestions:

I have three points of concern around the use of X-ray microCT as a proxy as applied here.

1. Sample size

The use of 8 shells at a 20cm resolution is a surprisingly small sample size from which to generate a record of [Co32-]. I understand that these analyses may be quite labor intensive, however, that alone cannot justify the use of such a small population. Especially without giving an indication of the amount of within-population variability or error of individual measurements. Results would be far more robust if the use of such a small sample size could be quantitatively justified and/or increased.

*I will note that it is not just the sample sizes of X-ray microCT samples that are quite small, but the use of only 5 *G. bulloides* for isotope analyses is also surprisingly small. Please see Fraass and Lowery (2017) for a thorough discussion of sample size, but where possible, closer to 20 shells is generally considered prudent.*

a. PS97/054-114 (Multiple core)

b. PS97/054-129 (Multiple core)

The reviewer is raising an important issue, which we attempted to explore more in depth by generating additional data and analyzing those. It is true that in many applications, a certain minimum number of specimens is needed to obtain a representative population mean. This number depends on the amount of variability that exists among the members of the population. Unlike the other proxies that reflect variations in sea surface conditions like as seasonal change, the CT-based proxy for dissolution should have a small variation in values, because all members of the population are exposed to the same conditions once they arrived on the seafloor. However, the reviewer is right that this assumption has not been sufficiently constrained by observations. Therefore, we carried out a test where we measured 30 tests of *G. bulloides* in two seafloor sediment samples and characterised their CT-based proxy individually. This is the highest number of specimens that can be plausibly analysed for this purpose at this moment given the technical limitations in sample throughput. This allowed us to characterise the variance in the CT-proxy values across the specimens and model the effect of pooling

multiple specimens, which represent an empirical, assumption-free determination of the distribution of the variance. The results show that our expectation of low variance was reasonable and both the mean and variance of a population containing 30 specimens can be in both samples approximated sufficiently (i.e., within the 95% confidence interval based on the 30 specimens), when only 8 specimens are measured, as was done for this study. We have now provided these additional analyses and the assessment of sample size effect in the Supplement and summarize the findings in the section “Sample size of the specimen from sediment sample” in Material and Method. Also, we reevaluated the data uncertainty due to sample size in the view of this new dataset and combined them with errors due to measurement and calibration.

As for the sample size in foraminiferal stable isotope analyses, though this is not a part of the main discussion, the reviewer is also right to point out that under some scenarios, using only five specimens may increase “scatter” in the resulting curve. To evaluate the effect of sample size on isotopic analyses, we used the existing five duplicate analyses from the same sediment samples. These reveal a scatter of about 0.15 ‰. This is larger than measurement uncertainty and therefore it is fair to say that the data contain scatter that could have been reduced, if we analyzed larger samples. However, the signal that we needed to detect is substantially larger (1 ‰) and sufficient to detect the pronounced negative peak of $\Delta\delta^{13}\text{C}$ planktic -benthic variation, which we mentioned in the manuscript. In the revised manuscript, we have now added this source of uncertainty to the other already considered precision of the mass spectrometer.

2. Selection of “unblemished” shells phrase may require some clarification, but I interpret this to

mean shells that appeared pristine under a light microscope. However, would this not bias selection towards shells that had different (lesser) degrees of dissolution? How degree of dissolution as measure by X-ray microCT may or may not correspond to appearance under a light microscope is an important missing piece of information!

We think “unblemished” may be a misleading word. What we mean is shells that are still intact, not fragmented or containing fractures and holes, or peeling of the surface, as well as specimens that are not obviously filled with sediments or overgrown by pyrite or other secondary minerals, which would all obscure the intrinsic signal of the shell calcite.

We changed and expanded the wording accordingly in the revised manuscript. Our sample selection criteria are as same as other studies of foraminiferal test weight measurement. Therefore, the bias due to sample selection is supposed to be as low as in conventional studies. We hope to be able to apply our dissolution proxy to fragmented shells or even any carbonate particle. However, we only know the dissolution process of the non-fragmented shells. Thus, we employed only non-fragmented shells in this study.

3. The claim of disassociation between shell thickness (surface processes) and density (bottom water).

a. The claim is made that while surface processes may impact SNW through shell thickness, X-ray microCT density measurements should be largely reflective of bottom water saturation. I think the distinction between density as a proxy for bottom water saturation and shell thickness for surface processes is broadly supported by both the authors' previous work and the latitudinal transect of core tops presented here. However, it strikes me that these two factors may not be truly independent. For example, a thinner shell may be more prone to dissolution if only due to an increase in surface area:volume of calcite. We know that CO₂ and thus near-surface [CO₃²⁻] likely changed over the deglaciation. Moreover, this region may (?) be within the influence of active upwelling along the Chile margin, which may have been variable in intensity and carbonate chemistry through the last deglaciation. Thus, how can one rule out or disentangle potential impacts of calcification (surface saturation) on subsequent dissolution (bottom water saturation)? This is not meant to be a challenge, but a genuine question I am left with.

The reviewer points out an essential aspect of the CT-proxy and rightly notes that there are situations under which surface and deep processes may be coupled. However, the argument that thinner shells are more prone to dissolution and thus affect the proxy values would be only valid under two conditions: the dissolution would lead to complete disappearance or damage of these shells and the remaining thicker shells would have intrinsically a different response to dissolution in the values of the CT-proxy. Otherwise, dissolution amount is only a function of shell surface, not surface/volume ratio, and as long as the shells have initially a similar ratio of the two different types of calcites that are the basis of the CT-proxy, the effect of partial dissolution should be the same on thin and thick shells. Nevertheless, to provide direct observational constraints on the process, we provide a new Supplemental Figure 3 b, in which we contrast a thin and a thick-walled specimen both found in the same sample. This shows that shells with different wall thickness were present in the samples and included in the analyses, and the extremely thin shells, where the CT-proxy could not be measured efficiently due to limited resolution of the CT scan. The result of SNW (shell weight)-thickness comparison (Supplement Figure 2) would indicate the influence from the population of extreme thin

shell in sample. On the other hand, under the processing of CT-proxy, such extreme thin shell can be identified. In this study, in order to remove the uncertainty caused by these shells, we did not include these types of shell in the calculation of CT-proxy. We have added a discussion of the above effects in the Methods section.

4. Novel proxy would be more robust with a supporting record

a. I think the greatest weakness here is really the remaining uncertainties around this proxy, some of which are articulated above. And, while I realize this would imply substantial additional work, I'd urge the authors to consider adding one or more supporting records utilizing a more conventional proxy. The obvious choice would be a cibicidoides (or other benthic foraminiferal) B/Ca (or d11B) record. Doing so would allow for a more robust direct comparison with the Atlantic records in question. And if carried out in contemporaneous samples, could provide a much-needed test of how this emerging density proxy functions downcore.

The main objective of this study is the application of the new proxy rather than proxy development. The novelty of this study is the $[\text{CO}_3^{2-}]$ reconstruction using a new proxy in a sediment core. Our proxy is particularly helpful where benthic foraminifera, required for B/Ca proxy, are too rare to facilitate geochemical analyses. This was the case in the studied sediment cores, which is why we are not able to provide a parallel B/Ca record. The reviewer is right that the CT-proxy is relatively new and its wider use would require additional verification. We are aware of the need for such tests but believe that these will be mainly justified when the proxy is applied outside the region where it has been initially developed. This is

precisely the reason why we constructed and tested the proxy in the SE Pacific, i.e., the same region where the studied sediment core comes from. We mentioned the potential limitation and highlighted the fact that the calibration and application in this manuscript occur in the same region in the revised manuscript.

We entirely agree with the reviewer that comparison with conventional proxy is crucial for developing our new methodology as a mature and more generally applicable proxy. Indeed, this is exactly the theme of our current research project, which is why we can share with the reviewer new unpublished results, which we believe go some way towards answering the raised criticism. In this dataset, we compared the B/Ca proxy and X-ray micro-CT proxy data in a series of identical sediment samples from the South Atlantic. As shown in the attached figure (Iwasaki et al., in prep.), the first direct comparison with the B/Ca proxy, which was possible because of more benthic foraminifera in

those samples, reveals that the two proxies correlated well, supporting the new proxy. We have every intention to continue developing our proxy for other regions and species and we are in the process of preparing the results shown in the figure above for publication. While we believe that a more detailed study of proxy validation benefits the community as a separate distinct paper, we leave the decision at the discretion of the editor whether the inclusion of the results shown here are indispensable for the potential acceptance of our manuscript for publication, in which case we would be willing to include these results in the supplement or main part of our paper.

In addition to this, I find interpretations of X-ray microCT ([Co32-]) data lacking in two areas.

1. Error bars associated with X-ray microCT based [Co32-] are shown in Figure 3 but not defined. What do these refer to? Do these include propagated error from both measurements and the calibration? It's a little hard to interpret without understanding what these bars represent, but I will note that most points seem to be "within error" of one another. If this is the case, it may be worthwhile to be more explicit (and cautious) in limiting interpretations to signals that appear truly robust. I also wonder if this issue could not be improved with an increase in sample size.

We thank the reviewer for pointing out the necessity to better explain the nature of the error bars. As described above, we have re-evaluated the error bars, also taking into account the effect of sample size and revised the description of Section "A new proxy method for [CO₃²⁻] applied to the SE Pacific".

2. Calculations of DIC are based on unrealistic assumptions.

Using modern conditions here does not seem appropriate. In addition to G-IG changes in TA as discussed, TA could potentially be influenced by both changes in surface production and export (e.g., associated with changes in carbonate export as discussed by the authors), as well as changes in dissolution. Temp and salinity are potentially more defensible but would require additional evidence that the water mass being recorded at each depth has been consistent. The same caveat would also apply to TA. If the authors are arguing for a change in influence of Northern (PDW) vs Southern sourced waters in some intervals, however, this becomes difficult.

I'd strongly suggest authors a) recalculate, propagating errors from a full range of relevant potential TA, T, and S values; b) seek additional constraints on parameters - perhaps using model results, external estimates of which I'm not aware, or re-evaluating their own data. For example, if authors collected B/Ca data, they may include Mg/Ca in their analytes which could further constrain temperature. OR c) acknowledge that sufficient constraints may not exist to calculate DIC and rely on [CO₃²⁻].

This is a very valid suggestion. As the reviewer pointed out, the carbonate system parameters we use are not necessarily consistent with the potential range across glacial-interglacial periods. Unfortunately, to provide a well quantified series of DIC variations would require a detailed knowledge of the ambient water alkalinity changes at that time, which is to our best knowledge unavailable. Since obtaining such data seems not possible, we decided to tackle this issue by showing [CO₃²⁻] variations rather than DIC in Figure 5a, because these parameters can be calculated directly from the CT-proxy.

In addition, we followed the reviewer's advice and recalculated the DIC change between the LGM (19 ka) and the early deglaciation (15 ka) using a range of temperature, salinity and alkalinity scenarios

taken from previous modelling and proxy-based studies. As a result, we are able to present the minimum values of DIC change as well as the potential range of realistic scenarios based on the tested range of assumption for the values of each parameter and conclude that under all tested scenarios, the deep-water mass in the SE Pacific would still have the potential to affect the global pCO₂ change during the last deglaciation. We have implemented the scarious and new calculations both in Figure 5a and in the extended paragraph from Line 404 onwards.

I've also added a few minor line-by-line points below.
- Could you label UCDW and LCDW on your Fig 1?

Following the suggestion, we revised the figure 1.

Line 83: There have been huge advances here in understanding the mechanistic relationships between B/Ca and carbonate ion in the past decade or so - I'd recommend the book Boron Proxies in Paleoceanography and Paleoclimatology (and of course references therein) as an overview.

Following the suggestion, we revised the sentence.

Line 89: The B/Ca ratio of epifaunal benthic foraminifera, particularly *Cibicidoides wuellerstorfi*, has so far been the most-often deep-water [CO₃²⁻] proxy, developed on the basis of empirical correlation with the deep-water carbonate saturation state ($\Delta[\text{CO}_3^{2-}]$)^{25, 30, 31}.

Line 135: Do you mean CaCO₃ saturation? Otherwise, why include the assumption of Ca consistency?

Because this sentence is redundant of Introduction, we deleted it and summarized in the above section as following.

Line 103: Assuming constant [Ca²⁺] on a time scale shorter than 100 ka³³, deep-water $\Delta[\text{CO}_3^{2-}]$ is primarily governed by [CO₃²⁻] on glacial-interglacial time scales.

Line 168: What is the test of significance used?

Because this increasing trend is a key of our results, we performed Student's t-test to proof the significant increase in [CO₃²⁻]. The sentence was revised as following.

Line 190: Thereafter, during the early deglaciation (15 – 19 ka BP), PCO₃ (LCDW) shows significant (Student's t-test; t = -15.2, df = 14, p < 0.01) [CO₃²⁻] increases from 75 to 95 $\mu\text{mol kg}^{-1}$, implying a marked DIC reduction in LCDW (Figure 3c).

Line 192: I'm assuming here and elsewhere this is a typo and you mean Ba/Ca (or possibly B/Ca?).

If this is not a typo, additional explanation of the utility of Br in this context is certainly warranted. In either case there is no discussion of sedimentary (I assume this is sedimentary and not from foraminifera?) Ba or Br analyses. This should be added to the methods.

“Br/Ca” is correct. We added the description about back ground of this proxy as follows.

Line 227: On the Chilean margin, variations in the bulk sediment Br/Ca ratios serve as an indication of the biological carbon pump efficiency, based on the fact that in this area sedimentary bromine (Br) and calcium (Ca) are closely associated with biogenic organic carbon and carbonate content, respectively⁴⁷.

Figure 3: Consider making the y axes comparable across these 4 plots as it is very difficult to compare magnitude of change at the different depths as currently plotted.

Following the suggestion, we revised the Figure 3.

Figure 3: Also, why do line plots continue past the last plotted point in b and c?

This is because there are hidden data points at the younger age. We revised the figure so that shows them.

Figure 3e&f: please include units.

They are dimensionless quantity.

Response to Reviewer #3:

(comments by the reviewer are printed in italics, our response in red)

General comments:

*The manuscript by Iwasaki et al. presents new data of X-ray Micro-Computer-Tomography of planktonic foraminifera *G. bulloides* to reconstruct deglacial changes of deep water CO₂ along a depth transect of the South-East Pacific. The method is recent and is applied for the first time for this area.*

The wide Pacific Ocean is not as well documented as the Atlantic Ocean and these new data are of interest. However, both results description and discussion are confusing and should be better organized.

Specific comments and suggestions:

While the paper aim at describing the oceanic circulation changes between the Pacific and Atlantic sector of the Southern Ocean, it does not take into account the fact that throughout the deglaciation, the cores studied might not be bathed by the same water masses, for example it might be the case for PC01 bathed by AAIW during different periods of the deglaciation as it has been published both for the Southwest and Southeast Pacific (Ronge et al., 2015; Haddam et al., 2020). The assumptions made on the Last Glacial Atlantic (AMOC) circulation could also be opposite to previous publications and this is not discussed (see comment Line 261-263). The authors compare their data with some published data from the Southwest, central and Southeast Pacific but some of the most recent publications are missing. The author should also compare their results with pH reconstructions for the deep waters of the Drake passage from $\delta^{11}\text{B}$ measured in deep sea corals (Rae et al., 2018 and Li et al., 2020) as they consider the influence of Pacific deep waters supplied to the Atlantic Ocean via the Drake Passage.

The referee is right and the same issue has been also pointed out by reviewer 2, who suggested a method of dealing with this uncertainty, which we followed. In the revised version, the discussion was rearranged based on the suggestions from both reviewers. Here we have tried to be more careful in describing the water mass distribution and AMOC variability after the LGM, in particular during the deglaciation. In addition, in the Figure 5 and Section “Pacific–Atlantic deep-water export through ACC transport:”, we added the comparison with the deep-water pH proxy obtained at the eastside of the Drake Passage, and discussed the impact of the deep-water supply from Pacific to Atlantic Ocean via Drake Passage on the water-mass distribution in the South Atlantic.

Other comments/questions:

Line 37: if you consider preindustrial interglacial, rounding up to 60 would be more appropriate

Following the suggestion, we corrected it.

Line 48: $\delta^{13}\text{C}$ and Cd/Ca indicates water masses geometry, not the strength of the flow. Like for air bubbles ice core CO₂ papers you could cite older publications (Curry et al., 1988 Duplessy et al., 1988,..)

Here we revised the manuscript as following:

Line 52: During the LGM, most paleoclimate proxy reconstructions indicate a generally shallower Atlantic MOC^{11, 12}. Overall, this slowed physical ocean movement resulted in increased nutrient and carbon concentrations in the deep ocean, particularly within glacial deep Pacific and Southern Ocean, resulting in isolation of these biogeochemical constituents from the surface ocean and atmosphere during the LGM¹¹⁻¹⁵.

Line 50 “in particular” why ? Deeper gradient have been observed for Atlantic and Indian Ocean also (for $\delta^{18}\text{O}$ for example).

The sentence was deleted from revised manuscript.

Line 55: could cite also Stephens and Keeling 2000, Ferrari et al., 2014...

Following the suggestion, we revised the manuscript.

Line 63: Following Talley 2013 the salinity minimum characterizing AAIW extend deeper than 1000 m, 1300-1400 m would clearly be more appropriate.

We revised this paragraph so that make it simple and easy to follow. The revised paragraph is following

Line 70: Today, in the subantarctic South Pacific around 45°S, the water mass structure deeper than 1000 m is broadly characterized by contributions of four principal water masses; Antarctic Intermediate Water (AAIW: ~1000 m), which flows northward, Pacific Deep Water (PDW: ~1000-2000 m) flow southwards, Antarctic Bottom Water (AABW: below ~4500 m), which moves northwards, and aged, modified North Atlantic Deep Water (NADW), which is transported by the Antarctic Circumpolar Current (ACC) from the Atlantic²⁹. The latter is hence commonly re-defined as Circumpolar Deep Water (CDW), and occupies a broad water depth range between 2000-4000 m depth, leading to varying degrees of mixing with the neighboring upper and lower water masses. As a result, modern CDW is divided into Upper CDW (UCDW), relatively more influenced by AAIW and PDW characteristics, and Lower CDW (LCDW), relatively more influenced by AABW and aged NADW characteristics (Figure 1).

Line 64: Mainly NADW? Following Talley 2013, the contribution to CDW are 9 Sv from the Pacific Ocean, 16 Sv from the Indian (from which 5Sv comes directly from NADW), and 13 Sv from the Atlantic, thus at maximum 18Sv from the Atlantic and 20 from the Indian and Pacific oceans.

As the reviewer pointed out, the sentence was incorrect. Please see the revised paragraph shown above.

Line 69: relatively more influenced by NADW and AABW characteristics

This sentence was revised following this suggestion.

Line 82: relationship between bottom water ΔCO_2 - and dissolution $\text{IDX}(\mu\text{CT})$ is also empirical

Yes, it's correct. Following the suggestion, the sentence was revised.

Line 89: The B/Ca ratio of epifaunal benthic foraminifera, particularly *Cibicidoides wuellerstorfi*, has

so far been the most-often deep-water $[\text{CO}_3^{2-}]$ proxy, developed on the basis of empirical correlation with the deep-water carbonate saturation state ($\Delta[\text{CO}_3^{2-}]$)^{25, 30, 31}.

Line 90: In Allen et al. 2019 that is cited, B/Ca has been measured for two cores (1627 m, 2541 m) that are influenced partly influenced by UCDW.

The sentence was revised as follow:

Line 92: For the Pacific sector of the Southern Ocean, B/Ca data from the subpolar Southwest Pacific show a significant decrease of $\sim 15 \mu\text{mol kg}^{-1}$ in $[\text{CO}_3^{2-}]$ within the UCDW during the LGM. This supposedly higher glacial DIC storage was followed by $\sim 20 \mu\text{mol kg}^{-1}$ increase during the early deglacial, indicating subsequent DIC release to the atmosphere²⁶.

Line 114: sections presenting CO32- concentrations would be nice also. AAIW would be better represent on a salinity section. As the cores are closer to the continent than the section represented, PC02 core is probably bathed by PDW and not CDW nowadays.

In order to show the $[\text{CO}_3^{2-}]$ distribution in the South Pacific, the bottom section image (Figure 1c) was changed into $[\text{CO}_3^{2-}]$. Our sample sites are located on the boundary of principal water masses that have changed actively with time. Thus, we consider that these sites are located within the CDW, which is mixture of surrounding water masses, in order to simplify the interpretation.

Line 137: temperature and salinity of deep water are clearly not stable throughout the deglaciation, would be nice that the authors estimate the error corresponding to changes in salinity and possible changes in temperature.

The effects of temperature and salinity variance between glacial-interglacial is small compared to analytical error of proxy, while we mentioned about this in the revised manuscript as following:

Line 160: The $[\text{CO}_3^{2-}]_{\text{sat}}$ is a constant depending on the bottom water temperature, salinity, and pressure at each core site. The overall effects of these parameters in the deep ocean $[\text{CO}_3^{2-}]_{\text{sat}}$ on glacial-interglacial time scales change about $0.5 \mu\text{mol kg}^{-1}$, based on $\sim 3^\circ\text{C}$ change in bottom water temperature, ~ 1.5 psu change in salinity, and ~ 120 m equivalent change in pressure⁴². Therefore, modern $[\text{CO}_3^{2-}]_{\text{sat}}$ values are employed to calculate down-core $[\text{CO}_3^{2-}]$. Based on the standard error of individual % Low-CT-number calcite volume measurement in each sample and our established calibration using multiple core samples, the uncertainty associated with reconstructing deep-water $[\text{CO}_3^{2-}]$ is $\sim 5.0 \mu\text{mol kg}^{-1}$ at the 95% confidence level. This makes it possible to detect $\sim 10 \mu\text{mol kg}^{-1}$ variations of $[\text{CO}_3^{2-}]$ on millennial or longer time-scales.

Line 145: Are both uncertainties linked to the empirical relationship and analytical

determination %Low CT number calcite volume both taken into account?

Yes, they include both errors. After re-considering accuracy of our data, description of data accuracy was revised.

Line 152: the shallowest site is bathed nowadays by PDW waters, and by AAIW during the warming events of the deglaciation (Haddam et al., 2020)

The changes in main water mass influencing PC02 was described in the section of “Transient zonal and meridional dynamics of Pacific deep-water carbonate chemistry” as follows:

Line 253: Thereafter, the water depth of ~1500 m close to PC01, was characterized by the effect of well-ventilated AAIW, and on the other hand, the water depth of ~2500 m, close to PC02, was characterized by the continuous influence of aged PDW during the early deglacial (Figure 4d)^{21, 48}.

Line 156: PC02 bathed by PDW nowadays, see previous comment.

The description was revised as follows:

Line 174: During the early deglaciation (15-19 ka BP), site PC01 (UCDW) yielded two [CO₃²⁻] peaks, one lower of 65 μmol kg⁻¹ at 16.5 ka, and another higher peak of 73 μmol kg⁻¹ at 15 ka.

Line 161-162: hard to follow: PC02 CO32- values from 19 to 17 ka indicate a sharp increase from 70 to ~90 μmol/kg, followed by a sharp decrease to values ~80 μmol/kg, with a rather stable period until 15 ka, followed by a decrease to 67 μmol/kg between 15 and 14 ka.

Based on the suggestion, we revised the description and figure as follow.

Line 184: Thereafter, the [CO₃²⁻] in PC02 (UCDW) showed stable values of ~80 μmol kg⁻¹ during the early deglaciation (17-19 ka BP), followed by a significant decrease to 67 μmol kg⁻¹ towards ~15 ka BP, implying a transient carbon-rich deep-water injection (Figure 3b).

Line 164: The first decrease in CO32- for core PC02 between 21 and 20 ka is not observed for PC03 but maybe the authors sentence “similar variations” is linked to values 82-75 μmol/kg of PC03 for the interval barren of foraminifera in core PC02?

As the reviewer pointed out, this sentence was confusing. Here we revised manuscript as follows:

Line 188: During the LGM (19-21 ka BP), PC03 (LCDW) shows values between 85-93 μmol kg⁻¹, while it shows lower values around 75 μmol kg⁻¹ at the end of the LGM (~ 19 ka BP).

Line 166-167: again difficult to follow: from 19 to 15 ka both cores indicates higher CO32-values

at the beginning (~90 μ mol/kg) and a decrease to lower values after. For the end of that period the different resolution makes it difficult to compare the two core records.

As the reviewer pointed out, the comparison between PC02 and PC03 is hard to understand. Now we focus on the explanation of [CO₃²⁻] change in PC03, and revised the sentences. Please see revised section of “Deglacial [CO₃²⁻] reconstruction: evolution of deep SE Pacific carbonate chemistry”

Line 173-178: long sentence, not very clear. What is the impact of the strong bottom current flow?

We revised this sentence in order to clarify our point. The proxy for bottom current changes is not concerning this discussion, thus we removed it. Instead, we now use the Ti/K ratio in the revised manuscript in order to support the potential occurrence of significant sediment deposition due to input of terrestrial materials from volcanic area in this area.

Line 196, fig 3e: PC02 indicate higher Br/Ca value at the end of the LGM than during 19-15 ka. For PC03 the values are more similar. It does not correspond to the authors description. Furthermore the comparison with Br/Ca of publication 40 might be unfounded: the authors from this publication argued that the high sedimentation rate of their core and the shallow depth favoured the preservation of organic matter (and Br) and carbonates and thus could be linked to the fluxes escaping the surface.

Following the suggestion, we revised the manuscript. The important point of this discussion is that decomposition of organic material in the sediment is not a principal factor of carbonate dissolution at these sites. This conclusion is based on the results that Br/Ca variation pattern does not correspond to carbonate dissolution pattern. The revised manuscript is following:

Line 231: In our study, there appears no apparent link between the variation in Br/Ca ratios and carbonate preservation at the bathyal sites PC02 and PC03 (Figure 3f). Although, local differences exist in surface productivity or depth-dependent differences in the carbonate and organic carbon preservation at our bathyal sites, the discrepancy between the Br/Ca ratio and carbonate dissolution patterns suggests that the organic carbon burial and decomposition in the sediment surface is not a principal controlling factor of carbonate dissolution intensity at our study sites. Therefore, we consider the carbonate chemical condition in deep-water mass to be principally reflecting a true water mass signal unaltered by porewater chemistry changes.

Line 220 “can be”?

The typo was corrected.

Line 242-246: The authors present PC01 as bathed by AAIW and UCDW on lines 152 and by PDW here. They do not discuss their reconstructed CO₂- record compared to Haddam et al, 2020 (and Ronge et al., 2015) indicating that during early deglaciation AAIW water occupied deeper depth

and notable bathed PC01. This changing water mass could correspond to the CO₃²⁻ increase observed from ~16.5 to 15 ka.

Thank you for the helpful suggestion. In the revised manuscript, we referenced the data of Haddam et al. (2020), and compared with it. The manuscript was revised as following.

Line 251: Previous studies using deep-water geochemical proxies on the Chilean margin suggested a strong stratification and effect of old PDW on water depths between 1500-2500 m at the end of the LGM (19 ka BP). Thereafter, the water depth of ~1500 m close to PC01, was characterized by the effect of well-ventilated AAIW, and on the other hand, the water depth of ~2500 m, close to PC02, was characterized by the continuous influence of aged PDW during the early deglacial (Figure 4d)^{21, 48}. The results of deep-water [CO₃²⁻] reconstructions in our study show an increase at Site PC01 and decrease at Site PC02 during the early deglaciation (15 – 16.5 ka BP), supporting earlier results and interpretations.

Lines 250-251: CO₃²⁻ record from core PS75/054-1 indicate variations from 85 to 100 μmol/kg that is not “largely invariant and rather of similar amplitude than variations from ~80 to 95 μmol/kg from core PC03.

As the reviewer pointed out, [CO₃²⁻] in PS75/054-1 gradually decreased. We suppose that this is caused by mixing with overlying LCDW. Based on this, we revised the manuscript as following:

Line 261: On the other hand, our deepest, abyssal site PS75/054-1, which showed a continuous decrease in the [CO₃²⁻] after the LGM, implies a strong effect of AABW supplied from the Ross Sea during the LGM. Thereafter, previous studies suggested the mixing with bathyal water masses from above during the early deglaciation based on εNd analyses²³.

Line 261-263: As explained before by the authors, GNAIW was shallower and a larger flux at shallow depth does not imply a larger influence on CDW during the last glacial maximum. As proposed by Keeling and Stephens, 2001, Ferrari et al., 2014 among others, the deep ocean circulation probably correspond to a deep overturning cell disconnected from the shallow overturning GNAIW cell.

Following the suggestions, we cited these studies and revised the description as follows.

Line 273: In the Atlantic Southern Ocean, on the other hand, shallower depth and a larger flux in the glacial intermediate water have been suggested to have caused the isolation from the atmosphere and shallow upper limit of underlying glacial NADW^{20, 49}.

Line 272-274: seems to be opposite to line 250-253

Here, we suggest the enhanced influence of AABW from below to the site PC02 and PC03 at the beginning of early deglaciation after LGM. Previous description at Line 250-253 was revised, and

they are consistent now.

Line 300 suppress “that”

We revised it.

Line 302-304: the authors could also comment on the fact that different tracers were used?

Here, we referenced the data of ventilation age reconstructions in the Pacific Southern Ocean and suggested that the proxy of deep-ocean circulation support the east-west imbalance in carbon storage variance. The revised manuscript is following:

Line 317: The ventilation age reconstructions in the Pacific Southern Ocean during the early deglaciation also revealed that the rejuvenation of deep-water mass was larger in the SW Pacific (from 3 to 1 kyr) than in the SE Pacific (from 1.5 to 1 kyr)^{21, 22}, which supports the east-west imbalance in Southern Pacific [CO₃²⁻] variations shown in our manuscript.

Line 317-330: main remark, the authors should consider deep sea corals $\delta^{11}B$ reconstructions from the Drake passage (Rae et al., 2018, Li et al., 2020).

We added the discussion to compare with the results of pH reconstruction at the opposite side of Drake Passage. Manuscript was revised as follows.

Line 335: On the downstream side of Drake Passage in the western South Atlantic, deep-water pH after the LGM was reconstructed using deep-sea coral boron isotope data (Figure 5c)^{54, 55}. These studies suggest that the deep-water pH significantly decreased at 15 ka BP at the sites within the upper overturning cell affected by Atlantic UCDW. On the other hand, a continuous increase during 15 – 18 ka BP was shown in the sites within the lower overturning cell affected by Atlantic LCDW^{54, 55}. These variations are consistent with [CO₃²⁻] variations in the Pacific UCDW (PC02) and LCDW (PC03) at the western, upstream side of the Drake Passage shown in our study. This suggests that the deep-water transport from the Pacific to the Atlantic contributed to the carbon redistribution during the early deglaciation, and was enhanced with the strengthening of Drake Passage through-flow⁵¹⁻⁵³.

Line 337-338: Productivity is thought to have been lower at the LGM in Antarctic surface waters, on the opposite, LGM productivity of subantarctic waters was enhanced, probably due to iron availability (Martin et al., 1990, Martinez-Garcia et al., 2011, 2014 among a number of other papers..)

Based on the suggestion, we revised the manuscript and text in Figure 6.

Line 354: During the LGM (Figure 6a), surface biological productivity in the subantarctic South Pacific is thought to have been generally high due to iron-rich conditions compared to interglacial periods^{56,57}.

Line 389-390: clearly not the good references for deep waters alkalinity changes.

The discussion of DIC calculation using assumed alkalinity was largely revised based on the way considering predicted Glacial-Interglacial variations in the deep-water parameter. Please see revised paragraph from Line 397.

Line 393 -394 suppress “also”

Corrected

Material and methods:

Supplemental fig 3a and 3b are referenced first in the text, corresponding to figure suppl 2a and 2b and fig. suppl. 1 is referenced after. Supplemental fig. 2c should be referenced line 450-453. The DIC calculations should be presented in the material and method part.

Based on the suggestion, we rearranged and reordered the Supplemental Figures.

Figures: it would help to have figures numbered

Figure 1: add c line 566 for 30°S section.

Corrected

Figure 4a: the authors should use the atmospheric $\delta^{13}\text{C}$ record of Bauska et al., 2016.

Following the suggestion, we change the data of atmospheric $\delta^{13}\text{C}$ into Bauska et al (2016).

Figure 5c: the vertical axis is wrong.

Corrected

Figure 6b: the carbonate counter pump will reduce the efficiency of the soft tissue pump, not “discharge CO₂ to the atmosphere”. Resumption of the Southern Ocean upwelling will conduct to ocean CO₂ degassing.

Based on the suggestion, we revised the text in Figure 6b. Also, we revised the description as follows:

Line 378: Secondly, at the end of the early deglaciation (15-16.5 ka BP) (Figure 6b), the biological pump efficiency was suppressed due to an increasing effect of the carbonate counter pump⁴⁷, and

enhanced upwelling resulting in carbon dioxide outgassing to the atmosphere.

REVIEWER COMMENTS

Reviewer #1 (Remarks to the Author):

I find that the authors revisions based on my review comments are robust and satisfactory. This version of the manuscript is more clearly articulated and well organized, with the benefit of additional and revised figures and supplemental information. I have no further comments on the existing manuscript and find it in an excellent state for publication. This new proxy tool using the emergent microCT technologies for paleoclimatology applications such as this one is an exciting advance in our field.

Reviewer #2 (Remarks to the Author):

Thank you for the consideration and very thorough response to my previous comments and those of other reviewers. The conclusions are more clearly supported in this revision. This is a very interesting manuscript that has exciting impact in understanding the role of the Southern Ocean in deglacial carbon storage.

I have a few very minor additional suggestions below:

100: remove “on the other hand”

102: “Assuming constant $[Ca^{2+}]$ on a time scale shorter than 100 ka³³, deep-water $\delta[CO_3^{2-}]$ is primarily governed by $[CO_3^{2-}]$ on glacial-interglacial time scales.”

I still think this should read “deep-water $\delta CaCO_3$ ”. As written the statement “ $\delta[CO_3^{2-}]$ is primarily governed by $[CO_3^{2-}]$ ” is correct but not very meaningful, and the assumption of constant $[Ca^{2+}]$ is not relevant. Perhaps I still am misunderstanding. If so, please clarify!

120: remove “previous”

141: “are changing” -> change

182: "the LGM"

221-224: this sentence is long and very complex. Consider breaking it into two for clarity.

238: mass -> masses. Maybe even better to be specific which water masses?

234-244: I don't quite understand this sentence. Maybe it's just missing an "or"?

244-246: I am not sure I quite follow this. Can you rephrase?

253-256: This is both important and really tough to follow. Can you rephrase and perhaps split into 2 sentences?

264: no "the"

318: "the rejuvenation of deep-water mass was larger"

This phrasing is a bit confusing or at least unfamiliar. Do you mean that deep-water masses in the SW Pacific were older/less ventilated?

357: no "relatively"

361: like -> likely OR remove "like" altogether

389: "re-juvenile" is not quite right. Maybe you mean "young" or "more ventilated"?

413: no "However"

415: No "a"

424-428: This is redundant with what has already been stated above for the LGM. Consider removing these lines and simply saying on line 423 "...temperature, salinity, and alkalinity ranged..."

Methods: This section probably needs one more close read through for clarity.

456-457: I don't quite understand. Does this mean 1 cm was sampled every 20 cm for a total of 10 samples?

460-461: I'd suggest rephrasing as "Only specimens that were not obviously filled with sediments or overgrown by pyrite or other secondary minerals were chosen."

464: Maybe remove "significantly thin" as this implies a test of significance was used.

511: Consider "followed" as opposed to "were"

528: "outermost chamber (final four chambers)" I find this confusing. Was only the final chamber or the final four analysed? Or perhaps something else and I completely misunderstand. Could you please clarify?

535: "previous"

535: no "have"

560-561: "Furthermore, we could not identify a distinct improvement in the standard analytical error under a sample size of 8 or more."

How was this tested or assessed? Or maybe there is a threshold of variance in standard error used (looking at Fig S2)

Fig 1 caption: "cores"

Fig 3 caption: "13C"

Fig 4 caption: "Eastern equatorial Pacific"

Reviewer #3 (Remarks to the Author):

The manuscript by Iwasaki et al. presents new data of X-ray Micro-Computer-Tomography of planktonic foraminifera *G. bulloides* to reconstruct deglacial changes of deep water CO₂ along a depth transect of the South-East Pacific. The authors improved a lot the manuscript taking account most of the suggestions of the different reviewers. The description of the new method, of the corresponding uncertainty is now clear and convincing. The authors compare their new [CO₂-] reconstructions with previous reconstruction in the South Pacific and South Atlantic thus increasing the interest of the paper.

The first part of the presentation of the results/discussion is still rather confused and should be reorganized. Furthermore, there are errors, with some figures not showing what the text says, wrong figure references, or plot shown that are not discussed/used in the text. This paper still needs some improvement before reaching the level of a Nature Communication paper.

I hope the following comment will help the authors. They already did a nice work in the first revision.

Line 70 to 81: This § is maybe not useful in the discussion. It is not linked with the previous or following § and could be associated to the cores position description that starts Line 172.

In those lines, the authors did not change the proposed depth for the different water masses according to the description of Talley et al. 2013. They might rely on other modern oceanographic reference (they could use also Macdonald et al., 2009 table 3 for example. I do not know any reference limiting AAIW at 1000 m in the SE Pacific ~45°S). The authors need to indicate their modern oceanographic reference.

The wording is not really adequate: "the water mass structure deeper than 1000 m... ; AAIW :~1000, PDW : ~1000-2000 m"

For the corresponding figure 1b, it would be nice to have the cross section showed close to the margin and not at 93°W, far from the cores (as there are a lot of salinity/oxygen/density data available). For figure 1c the authors used the real station data and not gridded ones, but why did they choose to show a longitudinal transect farther north at 30°S while in GLODAP 2.2 there is a transect close to their core at ~45°S (or transect at ~53°S from Glodap 1 would be closer than the transect at 30°S)?

Fig.2 legend: The lowest ΔCO_2 value is not at -13.4 ΔCO_2 but~-16.

Fig.3 : legend: f, add PCO₂

Line 171-208, looks like a result section. Not all the periods are described. Could be easier to have the §197-208 at the beginning, otherwise reading the description it is hard to understand why there is nothing written about those high values as if they do not exist. It is not Ti/K presented in Supplementary Fig. 1b but Zr/Rb. Where is Ti/K plot?

Line 175, instead of “two peaks”, the authors maybe wanted to present the range of the [CO₃2] values in that interval? Line 176: the value at 15ka does not correspond to the value indicated on the graph, ~70-71 μmol/kg. 73 μmol/kg is the value at 17ka.

Line 190 around 80 μmol/kg, one point at 75

Line 195: “continuous” not really; low resolution but oscillate between ~95 and 85 μmol/kg in the first part of the deglaciation.

Line 228-230: I do not think the CO₃2- reconstructed signal reflects the local remineralization of organic matter in surface sediments but I do not see the link with Br/Ca in the deep cores. Br is generally linked with the left-over organic matter that was not remineralized and is not indicative of the amount that reaches the sediments. Br/Ca in the shallow, high sedimentation rate core would be more convincing.

The large Δ¹³C difference between 17.5 and 16.5 could be linked to local enhanced productivity?

Line 244 replace “,” by “and”

Line 247: PC03 indicate an increase from 75 to ~90 μmol/kg from 19.5 to 17-14ka but its not what is observed for PC02.

Line 256-260 would need more explanations, it is not obvious that the new CO₃2- data support earlier interpretations. Nowadays AAIW has much higher values (80-100 μmol/kg) values than PDW. While an increase is indicated in PC01 values, it is similar to the increase seen in PDW (ODP1240). Thus the influence of AAIW on PC01 signal is not obvious. There is no decreasing trend in the PC03 data in the 16.5-15.5 ka interval.

Line 269-273, at ~20-19.5ka the three deep cores of this study indicate similar value, as if there was a strong vertical mixing at that time. It occurs again at ~17.5 ka. After “AABW” and “LCDW” indicate similar changes in CO₃2- values while UCDW has different behavior.

Line 276 and 279: In reference 25, GNAIW is considered at the LGM and not NADW and it does not extend to 3800 m water depth (the strong gradient is between 2500 to 3000 m depth). While the core MD07-3076Q, at 3800, m depth is considered under the influence of Pacific water in ref 25 (GPDW for Glacial Pacific Deep Water) and show effectively lower CO₃2- values during the LGM (60-80 μmol/kg) than the PC02 and PC03 values (77-93 μmol/kg), it is quite confusing that the author here consider it as NADW.

Line 314 rather missing before “than”?

Fig. 5c: pH coral data reference is missing.

Line 334 correct for Figure 5e

Line 361 correct “like” for likely?

Line 388-389, reference 54 should be indicated.

Line 389-394: From 15 to 12 ka a large CO₂- decrease is indicated for MD07-3076Q (fig. 5b) that does not fit with “increase in [CO₂] content in Atlantic CDW » and « reduction of carbon storage in the CDW as a whole »

Line 433 Why is Pacific LCDW limited to 135-150°W? Shouldn't it be from 180-170°W to 80-70°W? Depending on the limit chosen, the calculated volume could be very different.

Methods:

Line 535 correct “pervious” to previous

CO_3^{2-} (P,T,ALK,DIC) [$\mu\text{mol/kg}$]

Dear Editor and Reviewers:

In the following, we have compiled our specific replies to the reviewers' comments. The replies are arranged in a following manner:

1. *Reviewer's comments;*
2. **Our reply**
3. **Line No:** Actual sentence/phrase/paragraph we wrote and placed in text.

Reviewer #2 (Remarks to the Author):

Line100: remove "on the other hand"

Thank you for the correction. We revised the sentence.

Line102: "Assuming constant [Ca²⁺] on a time scale shorter than 100 ka³³, deep-water $\Delta[\text{CO}_3^{2-}]$ is primarily governed by [CO₃²⁻] on glacial-interglacial time scales." I still think this should read "deep-water ΔCaCO_3 ". As written the statement " $\Delta[\text{CO}_3^{2-}]$ is primarily governed by [CO₃²⁻]" is correct but not very meaningful, and the assumption of constant [Ca²⁺] is not relevant. Perhaps I still am misunderstanding. If so, please clarify!

The previous sentence was confusing. Here we mean that we can recognize the concentration of Ca²⁺ nearly constant, cause the residence time of Ca is significantly longer than the time scale of glacial-interglacial cycle. Here, we revised the sentence as follows.

Line 89: Deep-water $\Delta[\text{CO}_3^{2-}]$, with [Ca²⁺] and [CO₃²⁻] as variables, also largely control the dissolution of planktic foraminiferal tests, which are widely distributed in deep-sea sediments³². Because [Ca²⁺] in the ocean has a long residence time (~10⁶ years), it can be assumed to remain constant on a time scale shorter than 100 ka³³. Therefore, deep-water $\Delta[\text{CO}_3^{2-}]$ is primarily governed by [CO₃²⁻] on glacial-interglacial time scales.

120: remove “previous”

The sentence was revised.

141: “are changing” -> change

The sentence was revised.

182: “the LGM”

The sentence was revised.

221-224: this sentence is long and very complex. Consider breaking it into two for clarity.

Following the comments, we divided the sentences as follows.

Line 225: In our study, the results of the difference in stable carbon isotope between planktic and benthic foraminifera ($\Delta\delta^{13}\text{C}$ planktic-benthic), which represents the strength of bottom-surface ventilation²¹, suggested that ventilation have been poor at the LGM (19 ka BP) in bathyal PC03. Thereafter, it shows enhanced ventilation at the end of early deglaciation (15 ka BP) (Figure 3e).

238: mass -> masses. Maybe even better to be specific which water masses?

The sentence was revised as follows:

Line 242: Therefore, we consider the carbonate chemical condition in deep-water masses at the sites of our study (i.e., AAIW/PDW, CDW and AABW) to be principally reflecting a true water mass signal.

243-244: I don't quite understand this sentence. Maybe it's just missing an "or"?

Yes, we added "or" in the sentence.

244-246: I am not sure I quite follow this. Can you rephrase?

The sentence was revised as follow:

Line247: In theory, changes in deep water [CO_3^{2-}] at a given site can reflect either shifts in deep-water circulation or changes in the ambient water DIC content. In either case the deep-water [CO_3^{2-}] variation would imply a change in carbon storage.

253-256: This is both important and really tough to follow. Can you rephrase and perhaps split into 2 sentences?

As the reviewer pointed out, the description around here might be confusing. Here we revised the description as follows:

Line 254: A previous reconstruction of deep-water ventilation on the Chilean margin, using radiocarbon ventilation ages, suggested a strong stratification and effect of less ventilated water on water depths of ~1500 m close to PC01 at the end of the LGM (19 ka BP)²¹. Thereafter, during the early deglaciation (15 – 16.5 ka BP), the water mass close to PC01 was characterized by the effect of well-ventilated water (Figure 4d)²¹. In the SE Pacific, the deeper water mass ventilation has also been assessed based on bottom water oxygen concentration reconstruction using benthic foraminifera faunal assemblages⁵¹. Compared to results from a core at a shallower depth (~1600 m), the core at a depth of ~2500 m close to PC02 indicates a continuous influence of less ventilated deep water during the LGM to the early deglacial (Figure 4e)⁵¹. Our deep-water [CO_3^{2-}] reconstructions at Site PC02 show a decrease during the early deglaciation (15 – 16.5 ka BP), supporting the results of a continuous influence of less ventilated deep water (Figure 4b).

264: no "the"

The sentence was revised.

318: “the rejuvenation of deep-water mass was larger”

This phrasing is a bit confusing or at least unfamiliar. Do you mean that deep-water masses in the SW Pacific were older/less ventilated?

Here we intend that the reduction in deep water ventilation age is more significant in the Southwest than Southeast Pacific. In order to make these sentences clearer, we revised the description as follows.

Line 337: The ventilation age reconstructions in the Pacific Southern Ocean during the early deglaciation also revealed that the rejuvenation (i.e., reduction in ventilation ages) of deep-water mass was more pronounced in the SW Pacific (from 3 to 1 kyr) than in the SE Pacific (from 1.5 to 1 kyr)^{21, 22}, i.e., ventilation of the deep-water was more significantly enhanced in the SW Pacific during the early deglaciation.

357: no “relatively”

The sentence was revised.

361: like -> likely OR remove “like” altogether

The sentence was revised.

389: “re-juvenile” is not quite right. Maybe you mean “young” or “more ventilated”?

We changed “re-juvenile” into ‘young’.

413: no “However”

The sentence was revised.

415: No “a”

The sentence was revised.

424-428: This is redundant with what has already been stated above for the LGM. Consider removing these lines and simply saying on line 423 “...temperature, salinity, and alkalinity ranged...”

Following the suggestions, we revised the manuscript.

456-457: I don't quite understand. Does this mean 1 cm was sampled every 20 cm for a total of 10 samples?

In order to make this clear, the sentence was revised as follow:

Line 473: From the core of PS75/054-1, we took 1-cm thick samples every 20 cm interval and obtained ten samples in total.

460-461: I'd suggest rephrasing as “Only specimens that were not obviously filled with sediments or overgrown by pyrite or other secondary minerals were chosen.”

Following the suggestion, we revised the description.

464: Maybe remove “significantly thin” as this implies a test of significance was used.

Following the suggestion, we revised the description.

511: Consider “followed” as opposed to “were”

The sentence was revised based on the suggestion.

528: *“outermost chamber (final four chambers)” I find this confusing. Was only the final chamber or the final four analysed? Or perhaps something else and I completely misunderstand. Could you please clarify?*

We revised the sentence as follows:

Line 544: We also measured the test wall thickness of the four chambers in the final whorl of *G. bulloides* using iso-surface images. We measured the wall thickness at randomly selected five points in each chamber wall and showed the average thickness at a total of twenty points as the mean wall thickness.

535: *“previous”*

The sentence was revised.

535: *no “have”*

The sentence was revised.

560-561: *“Furthermore, we could not identify a distinct improvement in the standard analytical error under a sample size of 8 or more.”*

How was this tested or assessed? Or maybe there is a threshold of variance in standard error used (looking at Fig S2)

The student t-test was performed. Here we revised this sentence as follow:

Line 576: Furthermore, the two-tailed student t-test between sample size of 8 individuals and 30 individuals suggested no distinct difference between each population ($t = 0.64$, $df = 36$, $p > 0.05$).

Fig 1 caption: “cores”

We revised the figure captions.

Fig 3 caption: “ $\delta^{13}C$ ”

We revised the figure captions.

Fig 4 caption: “Eastern equatorial Pacific”

We revised the figure captions.

Reviewer #3 (Remarks to the Author):

Line 70 to 81: This § is maybe not useful in the discussion. It is not linked with the previous or following § and could be associated to the cores position description that starts Line 172.

In those lines, the authors did not change the proposed depth for the different water masses according to the description of Talley et al. 2013. They might rely on other modern oceanographic reference (they could use also Macdonald et al., 2009 table 3 for example. I do not know any reference limiting AAIW at 1000 m in the SE Pacific ~45°S). The authors need to indicate their modern oceanographic reference.

The wording is not really adequate: “the water mass structure deeper than 1000 m... ; AAIW :~1000, PDW : ~1000-2000 m”

For the corresponding figure 1b, it would be nice to have the cross section showed close to the margin and not at 93°W, far from the cores (as there are a lot of salinity/oxygen/density data available). For figure 1c the authors used the real station data and not gridded ones, but why did they choose to show a longitudinal transect farther north at 30°S while in GLODAP 2.2 there is a transect close to their core at ~45°S (or transect at ~53°S from Glodap 1 would be closer than the transect at 30°S)?

Thank you for the helpful suggestions. In accordance with these comments, we revised the manuscript as follows.

1. We moved the description about deep water mass structure to the chapter of Results and discussions from Line 178
2. We referred a new reference (Macdonald et al., 2009). Based on the deep-water mass definition using neutral density, we redefined the distribution of water mass in our study area.
3. Based on the suggestions, section lines in Figure 1 (Map) were revised.

The revised description is following:

Line 178: The deep-water structure in the SE Pacific around 45°S nearby our study sites is characterized by contributions of the following principal water masses (water mass

definition based on neutral density⁴⁶): Antarctic Intermediate Water (AAIW: ~500-1300 m), which flows northward, Pacific Deep Water (PDW: ~1300-2700 m) flowing southwards, Antarctic Bottom Water (AABW: below ~3800 m), which moves northwards. The rest is commonly defined as Circumpolar Deep Water (CDW) between ~2700-3800 m depth, leading to varying degrees of mixing with the neighboring upper and lower water masses. The modern CDW is divided into Upper CDW (UCDW), relatively more influenced by PDW characteristics, and Lower CDW (LCDW), relatively more influenced by AABW and aged modified North Atlantic Deep Water (NADW), which is transported by the ACC from the Atlantic²⁹ (Figure 1).

Fig.2 legend: The lowest ΔCO_3 - value is not at -13.4 ΔCO_3 but~-16.

The description might have been confusing. As for the selected samples in Figure 2a, we did not choose the sample from the site with lowest ΔCO_3 -. To avoid the misunderstanding, we revised the figure caption as follows

Line 942: Tests were obtained from the three core-top samples from depths of 1537, 2787 and 3851 m, and deep-water $\Delta[\text{CO}_3^{2-}]$ of 9.46, 2.42 and -13.42 $\mu\text{mol kg}^{-1}$, respectively.

Fig.3 : legend: f, add PC02

The figure caption was revised.

Line 171-208, looks like a result section. Not all the periods are described. Could be easier to have the §197-208 at the beginning, otherwise reading the description it is hard to understand why there is nothing written about those high values as if they do not exist. It is not Ti/K presented in Supplementary Fig. 1b but Zr/Rb. Where is Ti/K plot?

Based on this suggestion, we made a new chapter concerning high sedimentation rate interval before the main result chapter from Line 16. We also added the graph of Ti/K variation of PC02 and PC03 in the Supplemental Figure 1b.

Line 175, instead of “two peaks”, the authors maybe wanted to present the range of the [CO₃²⁻] values in that interval? Line 176: the value at 15ka does not correspond to the value indicated on the graph, ~70-71μmol/kg. 73μmol/kg is the value at 17ka.

That’s true, here we corrected the manuscript as follows:

Line 191: During the early deglaciation (15-19 ka BP), site PC01 (AAIW/PDW) showed the [CO₃²⁻] ranged from 65 to 73 μmol kg⁻¹.

Line 190 around 80μmol/kg, one point at 75

The number was revised.

Line 204: During the LGM (19-21 ka BP), PC03 (LCDW) shows values between 85-93 μmol kg⁻¹, while it shows lower values around 80 μmol kg⁻¹ at the end of the LGM (~ 19 ka BP).

Line 195: “continuous” not really; low resolution but oscillate between ~95 and 85 μmol/kg in the first part of the deglaciation.

The sentence was revised as follows:

Line 209: In contrast to our three shallower sites, it shows a decreasing [CO₃²⁻] trend from relatively high glacial values of 100 μmol kg⁻¹ towards Early Holocene values of 84 μmol kg⁻¹ (Figure 3d).

Line 228-230: I do not think the CO₃²⁻- reconstructed signal reflects the local remineralization of organic matter in surface sediments but I do not see the link with

Br/Ca in the deep cores. Br is generally linked with the left-over organic matter that was not remineralized and is not indicative of the amount that reaches the sediments. Br/Ca in the shallow, high sedimentation rate core would be more convincing. The large $\Delta\delta^{13}\text{C}$ difference between 17.5 and 16.5 could be linked to local enhanced productivity?

Referring these comments, we revised the description of this paragraph. We also mentioned about the data of $\Delta\delta^{13}\text{C}$. Based on the comparison with two proxies concerning biological pump efficiency, we conclude that carbonate dissolution is not principally controlled by organic carbon input. Revised paragraph is follows:

Line 230: On the other hand, remineralization of organic matter in the surface sediments may locally decrease porewater $[\text{CO}_3^{2-}]$ and thus induce carbonate dissolution. On the Chilean margin, variations in the bulk sediment Br/Ca ratios can serve as an indication of organic carbon delivery to the seafloor. This is because sedimentary bromine (Br) and calcium (Ca) are closely associated with biogenic organic carbon and carbonate content, respectively⁴⁸. Furthermore, the carbon isotope gradient represented by $\Delta\delta^{13}\text{C}$ planktic-benthic is also related to biological carbon pump efficiency^{49, 50}. Thus, high $\Delta\delta^{13}\text{C}$ planktic-benthic implies effective organic carbon transport from surface to deep ocean. In our study, there appears to be no consistent relationship between either proxy and carbonate preservation (Figure 3e, f). Thus, even if differences in productivity and/or organic carbon export existed at our bathyal sites, it is unlikely that the organic carbon burial and decomposition in the sediment surface was the principal controlling factor of carbonate dissolution intensity at our study sites. Therefore, we consider the carbonate chemical condition in deep-water masses at the sites of our study (i.e., AAIW/PDW, CDW and AABW) to be principally reflecting a true water mass signal.

Line 244 replace “,” by “and”

We revised the sentence.

Line 247: PC03 indicate an increase from 75 to $\sim 90\mu\text{mol/kg}$ from 19.5 to 17-14ka but its not what is observed for PC02.

Here, we intend that the $[\text{CO}_3^{2-}]$ at PC02 and PC03, are between the values of PC01 and PS75/054-. The sentences are revised as follows:

Line 249: In this study, the variations in $[\text{CO}_3^{2-}]$ at bathyal sites PC02 and PC03 fluctuate between the $[\text{CO}_3^{2-}]$ of the mesopelagic site PC01 and abyssal site PS75/054-1 (Figure 4b), implying that the deep-water masses at bathyal sites around 2000-3000 m water depth are principally a mixing product of the surrounding water masses.

Line 256-260 would need more explanations, it is not obvious that the new CO32- data support earlier interpretations. Nowadays AAIW has much higher values (80-100 $\mu\text{mol/kg}$) values than PDW. While an increase is indicated in PC01 values, it is similar to the increase seen in PDW (ODP1240). Thus the influence of AAIW on PC01 signal is not obvious. There is no decreasing trend in the PC03 data in the 16.5-15.5 ka interval.

As the reviewer pointed out, from the data at PC01, we cannot find the obvious effect of well-ventilated water (predicted to be high $[\text{CO}_3^{2-}]$). The first part of this paragraph was revised as follows:

Line 254: A previous reconstruction of deep-water ventilation on the Chilean margin, using radiocarbon ventilation ages, suggested a strong stratification and effect of less ventilated water on water depths of ~1500 m close to PC01 at the end of the LGM (19 ka BP)²¹. Thereafter, during the early deglaciation (15 – 16.5 ka BP), the water mass close to PC01 was characterized by the effect of well-ventilated water (Figure 4d)²¹. In the SE Pacific, the deeper water mass ventilation has also been assessed based on bottom water oxygen concentration reconstruction using benthic foraminifera faunal assemblages⁵¹. Compared to results from a core at a shallower depth (~1600 m), the core at a depth of ~2500 m close to PC02 indicates a continuous influence of less ventilated deep water during the LGM to the early deglacial (Figure 4e)⁵¹. Our deep-water $[\text{CO}_3^{2-}]$ reconstructions at Site PC02 show a decrease during the early deglaciation (15 – 16.5 ka BP), supporting the results of a continuous influence of less ventilated deep water (Figure 4b). On the other hand, the $[\text{CO}_3^{2-}]$ at Site PC01 was similar to PC02 at the end of early deglaciation (15 ka BP), which appears to contradict the results of existing ventilation age reconstructions²¹. In this context, it is important to highlight that ventilation age is an indirect proxy of carbonate chemistry. In theory, the ventilation age results could be

reconciled with our results if the chemical properties of the studied water mass were significantly different and its carbonate chemistry was primarily related to its ventilation age. However, the low $[\text{CO}_3^{2-}]$ reconstruction obtained for Site PC01 with our method implies that the water mass at Site PC01 was located at the boundary of AAIW and PDW. Therefore, it should have been affected by aged PDW during the early deglacial at least as strongly as at present, which is not consistent with observed large differences in ventilation ages²¹.

Line 269-273, at ~20-19.5ka the three deep cores of this study indicate similar value, as if there was a strong vertical mixing at that time. It occurs again at ~17.5 ka. After “AABW” and “LCDW” indicate similar changes in CO₃²⁻ values while UCDW has different behavior.

Here, we'd like to say about the time period at the end of LGM (19 ka BP). In order to make it clearer we revised this paragraph.

Line 276 and 279: In reference 25, GNAIW is considered at the LGM and not NADW and it does not extend to 3800 m water depth (the strong gradient is between 2500 to 3000 m depth). While the core MD07-3076Q, at 3800, m depth is considered under the influence of Pacific water in ref 25 (GPDW for Glacial Pacific Deep Water) and show effectively lower CO₃²⁻ values during the LGM (60-80 $\mu\text{mol/kg}$) than the PC02 and PC03 values (77-93 $\mu\text{mol/kg}$), it is quite confusing that the author here consider it as NADW.

Thank you for the suggestion. We revised the description regarding these points. In the revised manuscript, we summarized the structure of the deep-water stratification and isolation of the glacial CDW based on the results of this study and previous studies as follows.

Line 286: At the end of the LGM (19 ka BP), our bathyal $[\text{CO}_3^{2-}]$ reconstructions from sites PC03 differ significantly from values at ODP1240 (PDW) and PS75/054-1 (AABW). Considering that the South Pacific deep-water mass structure was supposedly strongly stratified during this period (Figure 4d, f)^{21, 23}, glacial CDW should have been less influenced by the overlying PDW supplied from north and the underlying AABW

supplied from south. A glacial CDW isolated from surrounding water masses was also inferred for the Atlantic Southern Ocean, and explained by a shallower depth and a higher flux of glacial intermediate water^{20, 52}. Specifically, reconstructions in the subantarctic Southern Atlantic revealed that the Atlantic CDW at the depth of ~3800 m showed lower [CO₃²⁻] (~70 μmol kg⁻¹) than underlying Atlantic AABW (~90 μmol kg⁻¹) at the depth of ~5000 m²⁵(Figure 5b), which suggest the isolation of glacial Atlantic CDW from the other water masses. Such Atlantic CDW, which partly originated from glacial NADW, may have been channeled into the South Pacific by the ACC. Furthermore, glacial Pacific CDW may have been re-supplied into Atlantic CDW via the Drake Passage.

Line 314 rather missing before “than”?

The sentence was revised.

Fig. 5c: pH coral data reference is missing.

The figure caption was revised.

Line 334 correct for Figure 5e

The sentence was revised.

Line 361 correct “like” for likely?

The sentence was revised.

Line 388-389, reference 54 should be indicated.

The reference was added.

Line 389-394: From 15 to 12 ka a large CO₃²⁻ decrease is indicated for MD07-3076Q (fig. 5b) that does not fit with “increase in [CO₃²⁻] content in Atlantic CDW » and « reduction of carbon storage in the CDW as a whole »

The previous description was wrong. It is suggesting about the time period of the early deglaciation (15-19 ka) not after 15 ka. Here we revised the manuscript as follows:

Line 411: In addition, young and carbon-poor waters originating from upwelled northern-sourced PDW were transported into Atlantic CDW, which may have contributed to the increase in [CO₃²⁻] content in Atlantic CDW. This suggests that better ventilated (i.e., low DIC) deep-water was exported to the Atlantic and mixed into Atlantic LCDW and AABW, contributing to the reduction of carbon storage in the CDW as a whole during the early deglaciation (15 – 19 ka BP), (Figure 6b).

Line 433 Why is Pacific LCDW limited to 135-150°W? Shouldn't it be from 180-170°W to 80-70°W? Depending on the limit chosen, the calculated volume could be very different.

The description of assumed CDW range was incorrect. Now we revised the sentence. Based on the revised distribution range, we re-calculated the carbon efflux. The revised sentences are as follow:

Line 448: If deep-water DIC of LCDW in the South Pacific (assumed water mass distribution: 30°S-65°S, 75°W-180°W, 3000-4000 m water depth, with reference to the modern water mass distribution) uniformly was reduced by $\sim 35 \pm 26 \mu\text{mol kg}^{-1}$ as a minimum estimated value, the amount of carbon efflux from this water mass can be calculated as $\sim 9.4 \pm 7.5 \text{ GtC}$. This amount would correspond to $\sim 15 \pm 11 \%$ of the total carbon emission to the atmosphere. Although our DIC and carbon efflux calculations broadly vary with the assumed values of deep ocean alkalinity and volume of target water mass, we suggest that carbon storage in South Pacific LCDW played a critical role in the atmospheric $p\text{CO}_2$ rise during the early deglaciation.

Methods:

Line 535 correct “pervious” to previous

Thank you. We revised the sentence.